**Brief Communication**

# Harnessing clonal gametes in hybrid crops to engineer polyploid genomes

Yazhong Wang [1], Roven Rommel Fuentes [1], Willem M. J. van Rengs[1], Sieglinde Effgen[1], Mohd Waznul Adly Mohd Zaidan[1], Rainer Franzen[2], Tamara Susanto [1], Joiselle Blanche Fernandes [1], Raphael Mercier [1] & Charles J. Underwood [1] ✉

Heterosis boosts crop yield; however, harnessing additional progressive heterosis in polyploids is challenging for breeders. We bioengineered a 'mitosis instead of meiosis' (*MiMe*) system that generates unreduced, clonal gametes in three hybrid tomato genotypes and used it to establish polyploid genome design. Through the hybridization of *MiMe* hybrids, we generated '4-haplotype' plants that encompassed the complete genetics of their four inbred grandparents, providing a blueprint for exploiting polyploidy in crops.

Heterosis, or hybrid vigor, describes the increased yield and robustness of hybrid plants relative to their parents and is a cornerstone of modern crop breeding[1]. Beyond biparental heterosis, autopolyploid progressive heterosis (APH) is observed in maize, potato and alfalfa when genomic segments from four distinct grandparents are combined, leading to additional heterotic effects[2]. APH has yet to be fully exploited in commercial breeding because meiosis reassorts genotypes and genetically uniform seeds that benefit from APH cannot be produced. The 'mitosis instead of meiosis' (*MiMe*) system previously established in *Arabidopsis* and rice leads to clonal, unreduced gametes[3–7], but has yet to be established in a dicot crop or tested in the engineering of polyploid genomes by design. Here, we established polyploid genome design in tomato to allow the controlled combination of four predefined genome haplotypes through the hybridization of clonal gametes produced by two distinct hybrid parents.

We set out to establish a *MiMe* system in tomato to produce clonal gametes in a controlled manner. Building on fundamental insights from tomato meiotic mutants (Supplementary Note 1), we found that a functional *MiMe* system could be established in inbred tomato through mutation of *SlSPO11-1*, *SlREC8* and *SlTAM* (Fig. 1a−c, Extended Data Figs. 1 and 2, Supplementary Figs. 1–16 and Supplementary Tables 1–4). We implemented the *MiMe* system in three hybrid tomato genotypes, including the Moneyberg-TMV × Micro-Tom (MbTMV-MT) model hybrid, the date-tomato commercial hybrid 'Funtelle' and the truss tomato commercial hybrid 'Maxeza' (Fig. 1a−c). We identified two independent MbTMV-MT, three independent Funtelle and three independent Maxeza lines with biallelic mutations in *SlSPO11-1*, *SlREC8* and

*SlTAM* (Supplementary Table 4). We focused on one putative hybrid *MiMe* line per hybrid and found that all were capable of producing unreduced pollen (Fig. 1c). We prepared chromosome spreads from these three hybrid *MiMe* lines for cytological analysis of meiosis (Fig. 1d and Supplementary Fig. 17). In wild-type meiosis, homologous chromosomes synapse and form 12 highly condensed recombining bivalents, which is followed by two rounds of segregation to generate tetrads (Fig. 1d). In contrast, the hybrid *MiMe* mutants went through a mitotic-like cell division in which 24 univalents were identifiable at diakinesis, followed by the production of dyads with a single round of segregation (Fig. 1d and Supplementary Fig. 17). Compared with the MbTMV-MT fruits (20.22 ± 2.04 g), the MbTMV-MT *MiMe* mutants produced smaller fruits (11.31 ± 0.74 g) that contained fewer seeds (Fig. 1e,f). However, these seeds were larger than wild-type seeds and gave rise to tetraploid offspring at high penetrance (93%) (Fig. 1g,h and Supplementary Fig. 18). We sequenced tetraploid offspring from two MbTMV-MT[MiMe] parents (six offspring from each) and controls and used polymorphic genetic markers between the parental genomes (MbTMV and MT) to infer crossover recombination (Supplementary Fig. 19). As expected, F₁ plants were heterozygous across the genome (0.5 allele frequency), whereas F₂ plants showed divergence to homozygous states (0 and 1 allele frequency), indicating that crossovers had occurred (Fig. 1i). In contrast, tetraploid *MiMe* offspring (0.5 allele frequency) demonstrated a pattern similar to that of the F₁ hybrid controls without crossovers (Fig. 1i). In some *MiMe* offspring (3/12), we observed local deviations from the 0.5 allele frequency (Fig. 1i) and normalized read coverage, indicating chromosome instability (Supplementary

[1]Department of Chromosome Biology, Max Planck Institute for Plant Breeding Research, Cologne, Germany. [2]Central Microscopy (CeMic), Max Planck Institute for Plant Breeding Research, Cologne, Germany. ✉e-mail: cunderwood@mpipz.mpg.de

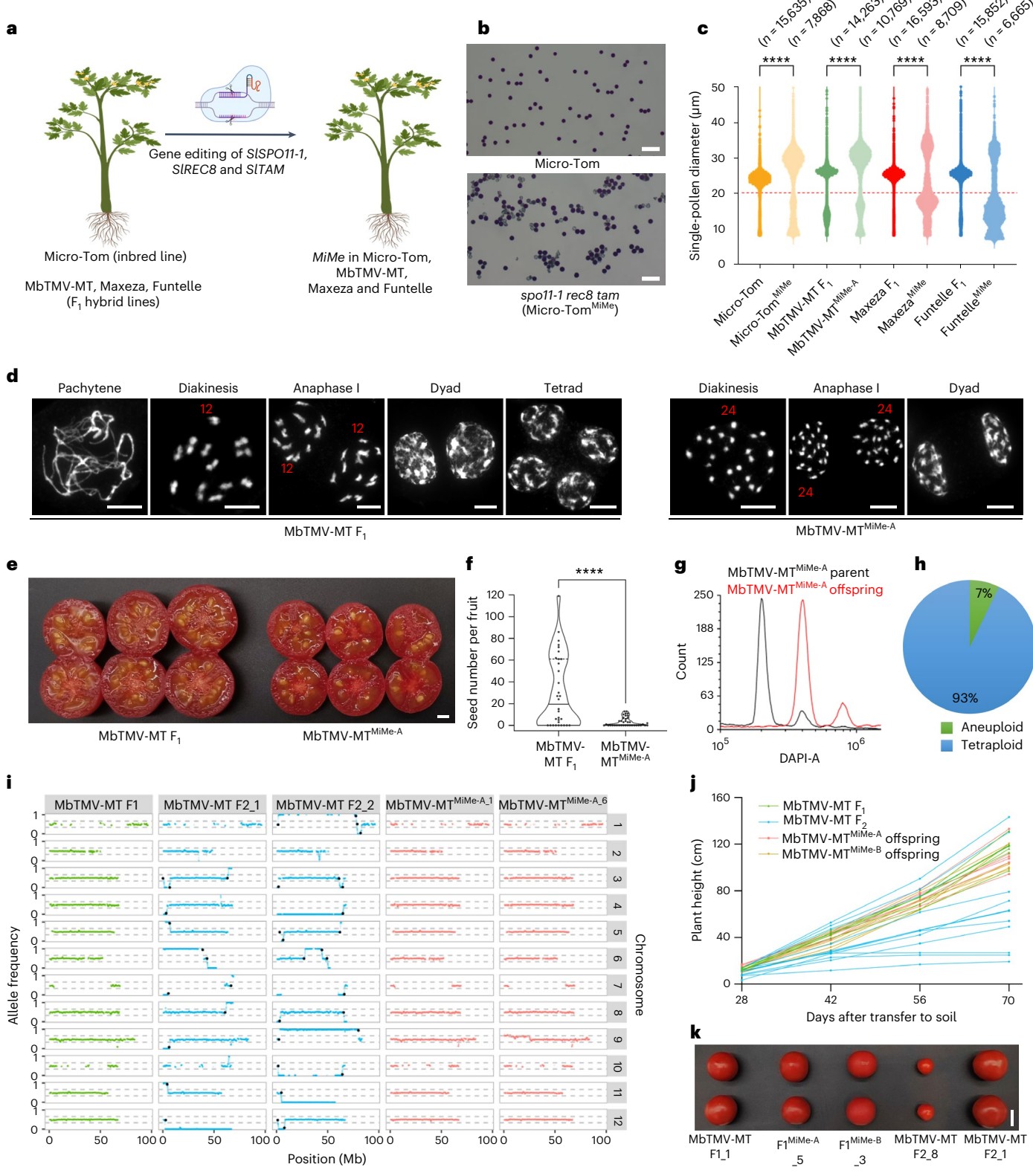

Fig. 20). In one of the offspring (MbTMV-MT$^{MiMe-A}$_6), we validated the partial loss of one of the MbTMV copies of chromosome 9 (Fig. 1i). These chromosome truncations may arise due to SPO11-1-independent DNA double-strand breaks (for example, arising from DNA replication stress or environmental DNA damage) that could not be repaired by homologous recombination due to the absence of homologous chromosome pairing and disturbed sister chromatid cohesion. Next, we

explored the phenotypic behavior of hybrid *MiMe* offspring and control F$_1$ and F$_2$ plants, focusing on plant height, fruit and seed development and leaf morphology (Fig. 1j,k and Supplementary Fig. 21). We found highly diverse phenotypes among the MbTMV-MT F$_2$ offspring, whereas highly consistent phenotypes were observed among the F$_1$ hybrid and *MiMe* tetraploid offspring (Fig. 1j,k and Supplementary Fig. 21). These findings illustrate that unreduced, clonal gametes can be produced

**Fig. 1 | Clonal gametes by *MiMe* lead to the inheritance of genome-wide heterozygosity and plant phenotypes in tomato. a**, Schematic of establishing *MiMe* in four tomato genotypes. Created with Biorender.com. **b**, Alexander staining of pollen from Micro-Tom ($n = 32$) and *spo11-1 rec8 tam* ($n = 83$). Scale bars, 100 µm. **c**, Single-pollen diameter in wild type and *MiMe* plants. Pollen grains below the red dashed line are deemed as nonviable pollen grains. *P* values were calculated using the Wilcoxon rank-sum test: ****$P < 0.0001$. See the 'Statistics and reproducibility' section in the Methods for more information. **d**, Chromosome behavior of male meiocytes in the wild type and MbTMV-MT$^{MiMe-A}$. Wild type: pachytene ($n = 53$), diakinesis ($n = 46$), anaphase I ($n = 35$), dyad ($n = 28$), tetrad ($n = 48$); MbTMV-MT$^{MiMe-A}$: diakinesis ($n = 47$), anaphase I ($n = 56$), dyad ($n = 57$). Scale bars, 10 µm. **e**, Transverse anatomical view of MbTMV-MT F$_1$ and hybrid MbTMV-MT$^{MiMe-A}$ fruits. Scale bar, 1 cm. **f**, Seed number per fruit in the wild type ($n = 74$) and MbTMV-MT$^{MiMe-A}$ ($n = 75$). The median is shown by a solid black line and quartiles are shown by dashed black lines. The *P* value was calculated using an unpaired two-tailed *t*-test: ****$P < 0.0001$, with an exact *P* value of $P = 4.99 \times 10^{-5}$. **g**, Flow cytometry analysis of diploid parent MbTMV-MT$^{MiMe-A}$ (black) and tetraploid MbTMV-MT$^{MiMe-A}$ (red) offspring. **h**, The ploidy level of MbTMV-MT$^{MiMe-A}$ offspring estimated by flow cytometry of leaf nuclei. Tetraploid plants ($n = 77$) were validated and six plants were potentially aneuploid. **i**, Whole-genome sequencing and allele frequency analysis of an MbTMV-MT F$_1$ plant, two MbTMV-MT F$_2$ plants and two tetraploid MbTMV-MT$^{MiMe-A}$ offspring. The allele frequency distribution between MbTMV (0 on the *y* axis) and Micro-Tom (1 on the *y* axis) is shown. Meiotic crossover positions are approximated with black dots. **j**, Time series of plant height of MbTMV-MT F$_1$ (green), MbTMV-MT F$_2$ (blue) and two MbTMV-MT$^{MiMe}$ offspring populations (red and brown). **k**, Fruit morphology of MbTMV-MT F$_1$, MbTMV-MT$^{MiMe-A}$ offspring, MbTMV-MT$^{MiMe-B}$ offspring and two different MbTMV-MT F$_2$ plants. Scale bar, 2 cm.

in hybrid tomato plants containing mutations in the *MiMe* genes and that their tetraploid offspring maintain genome-wide heterozygosity and plant characters.

Next, we harnessed the *MiMe* system for polyploid genome design to generate plants that contained the complete genetic repertoire of their four inbred grandparents. According to classical nomenclature, such plants could be referred to as 'nonrecombinant, double-cross hybrids' but, for simplicity, we refer to them as '4-haplotype' (4-Hap) plants[2,8]. To generate 4-Hap plants, we designed two sets of crosses between hybrid *MiMe* plants (MbTMV-MT$^{MiMe}$ × Maxeza$^{MiMe}$; MbTMV-MT$^{MiMe}$ × Funtelle$^{MiMe}$) (Fig. 2a and Supplementary Table 8). We used a platinum-grade genome assembly of the inbred grandparent MbTMV[9], generated a platinum-grade assembly of Micro-Tom and developed haplotype-resolved assemblies of the hybrid parental lines (Funtelle and Maxeza) to identify unique single-nucleotide polymorphisms (SNPs) for each haplotype in each cross (Supplementary Note 2, Supplementary Figs. 22–27 and Supplementary Tables 5 and 6). Next, we aimed to fully characterize the distinct haplotypes in a set of 18 putative 4-Hap plants (13 from MbTMV-MT-Maxeza$^{MiMe}$ and 5 from MbTMV-MT-Funtelle$^{MiMe}$) using whole-genome sequencing. Using the haplotype-specific SNP markers, the presence of all four parental genomes in each 4-Hap plant was validated (Fig. 2b,c). In addition, we found that each 4-Hap plant inherited mutations in the *MiMe* target genes, which confirmed genetic inheritance from both parents (Supplementary Table 4). Furthermore, cytological analysis of a subset of lines demonstrated that 4-Hap plants had the expected 48 chromosomes (Fig. 2d and Supplementary Fig. 28). By examining allele frequency, we confirmed equal dosage from both parents and tetraploids in 16 of 18 4-Hap plants analyzed (Supplementary Figs. 31–35). Chromosome truncation was observed in MbTMV-MT-Maxeza$^{MiMe-9}$ (Supplementary Fig. 30). We predicted the agronomically relevant gene dosage in 4-Hap plants based on parental genome sequences and

subsequently counted the allele frequency in the 4-Hap plants themselves (Fig. 2e). This revealed that genotypes were inherited as expected if gametes were clonal (Fig. 2e). Synteny analysis of the Funtelle haplotypes and annotation with a *Meloidogyne incognita* (*Mi-1*) resistance gene marker confirmed introgression in the structurally divergent Funtelle haplotype 1 (Fig. 2f and Supplementary Fig. 32), which was further elaborated by elevated numbers of *Solanum peruvianum* (*Mi-1* donor) genes along that haplotype (Fig. 2g). The genotyping analysis showed that MbTMV-MT-Maxeza$^{MiMe}$ plants had more copies of the *Tm-2²* haplotype than MbTMV-MT-Funtelle$^{MiMe}$ plants (Fig. 2e and Supplementary Fig. 32), as predicted from the parental genome sequences and the distribution of *S. peruvianum* (*Tm-2²* donor) genes (Fig. 2g and Supplementary Fig. 33). Remarkably, phenotyping tests of 4-Hap plants demonstrated normal vegetative growth and the production of well-organized inflorescences that harbor seedless fruits, as well as higher chlorophyll content in mature leaves (Fig. 2h,i and Supplementary Figs. 30, 31 and 36). In summary, we found that 4-Hap plants contain four genomes directly descended from their four inbred grandparents and allow a unique combination of plant characters.

Polyploid genome design has the potential to control genetic heterozygosity in polyploids, thereby allowing APH to be fully exploited in agriculture. In this report, we demonstrate that clonal gamete production in hybrid crop genotypes allows precise polyploid genome engineering; however, the exploitation of APH will involve further steps. This will require the development of four-way heterotic groups, which could be driven by using genomic selection to identify higher-order combining abilities between grandparental lines[10]. Tetraploidy doubles the length of the genetic map of a diploid crop, meaning that breeders could incorporate more unique characteristics in elite lines that were previously abandoned because of polygenic inheritance or prohibitive linkage drag. For example, our blueprint could facilitate the introgression of one or multiple complete 'wild' genomes into cultivated crops

**Fig. 2 | Precise engineering of tetraploid plants with four nonrecombined haplotypes via polyploid genome design. a**, Schematic of generation of tetraploid 4-Hap plants that contain the complete genetic repertoire of their four inbred grandparents. Created with BioRender.com. **b,c**, Presence of haplotype-specific markers in 13 MbTMV-MT$^{MiMe}$ × Maxeza$^{MiMe}$ offspring, 2 Maxeza F$_1$ plants, 5 MbTMV-MT$^{MiMe}$ × Funtelle$^{MiMe}$ offspring, 2 Funtelle F$_1$ plants and 3 MbTMV-MT F$_1$ plants. Colors represent each haplotype tested and the size of the circle represents the percentage of markers found (a circle completely filling the square denotes that 100% of markers were found). **d**, Chromosome spreads of male meiocytes from a tetraploid MbTMV-MT$^{MiMe}$ × Funtelle$^{MiMe}$ offspring 4-Hap plant at the diakinesis stage ($n = 36$). Scale bar, 10 µm. **e**, The expected gene dosage (histogram) and whole-genome sequencing-based genotyping (box-and-whisker plot; solid black line is the median, boxes show quartiles and whiskers show values within 1.5× the interquartile range above and below the quartiles) of MbTMV-MT-Funtelle$^{MiMe}$ ($n = 5$) and MbTMV-MT-Maxeza$^{MiMe}$ ($n = 13$) 4-Hap plants. The genes tested encode tobacco mosaic virus resistance (*Tm-2²*, Solyc09g018220), *M. incognita* resistance (*Mi*, Solyc06g008720), self-pruning (*SP*, Solyc06g074350), dwarfism (*D*, Solyc02g089160) and *Fusarium*

wilt resistance (*I*, Solyc11g011180). **f**, Genomic rearrangements on chromosome 6 at the *Mi-1* resistance locus. The haplotypes are depicted as horizontal lines and are (from top to bottom) MbTMV, Micro-Tom and Funtelle-2 and Funtelle-1 haplotypes. **g**, Frequency of genes derived from *S. peruvianum* (donor of *Tm-2²* and *Mi-1*) per genome haplotype. **h**, Images of a tetraploid MbTMV-MT$^{MiMe}$ × Funtelle$^{MiMe}$ offspring 4-Hap plant. Whole-plant morphology (left), structure of a branch with ripening fruits (middle), fully ripened fruits (top right), harvested fruits (middle right) and cut fruits (bottom right) are shown. Scale bars, 2.5 cm. **i**, Left, single-fruit weights of MbTMV-MT F$_1$ ($n = 40$), Funtelle F$_1$ ($n = 34$) and four tetraploid MbTMV-MT-Funtelle$^{MiMe}$ plants (1, $n = 41$; 2, $n = 53$; 3, $n = 46$; 5, $n = 52$). Right, single-fruit weights of MbTMV-MT F$_1$ ($n = 40$), Maxeza F$_1$ ($n = 17$) and 12 tetraploid MbTMV-MT-Maxeza$^{MiMe}$ plants (1, $n = 27$; 2, $n = 31$; 3, $n = 36$; 4, $n = 18$; 5, $n = 35$; 6, $n = 28$; 8, $n = 11$; 9, $n = 23$; 10, $n = 15$; 11, $n = 33$; 12, $n = 30$; 13, $n = 25$). The same MbTMV-MT F$_1$ data are used twice in this panel for comparison. We used ordinary one-way ANOVA followed by Šídák's multiple-comparison test: $P < 0.05$, 'NS' (not significant); *$P < 0.05$; **$P < 0.01$; ***$P < 0.001$; ****$P < 0.0001$. See the 'Statistics and reproducibility' section (Methods) for more information.

to facilitate abiotic and biotic stress resistance encoded by several unlinked genes[11–13]. It has not escaped our attention that polyploid genome design has major implications for hybrid potato breeding, as it provides the flexibility to perform genetic improvement at the diploid level and then harness heterosis at the polyploid level on the farm[14–16]. Taking a wider perspective, polyploid genome design could be used for

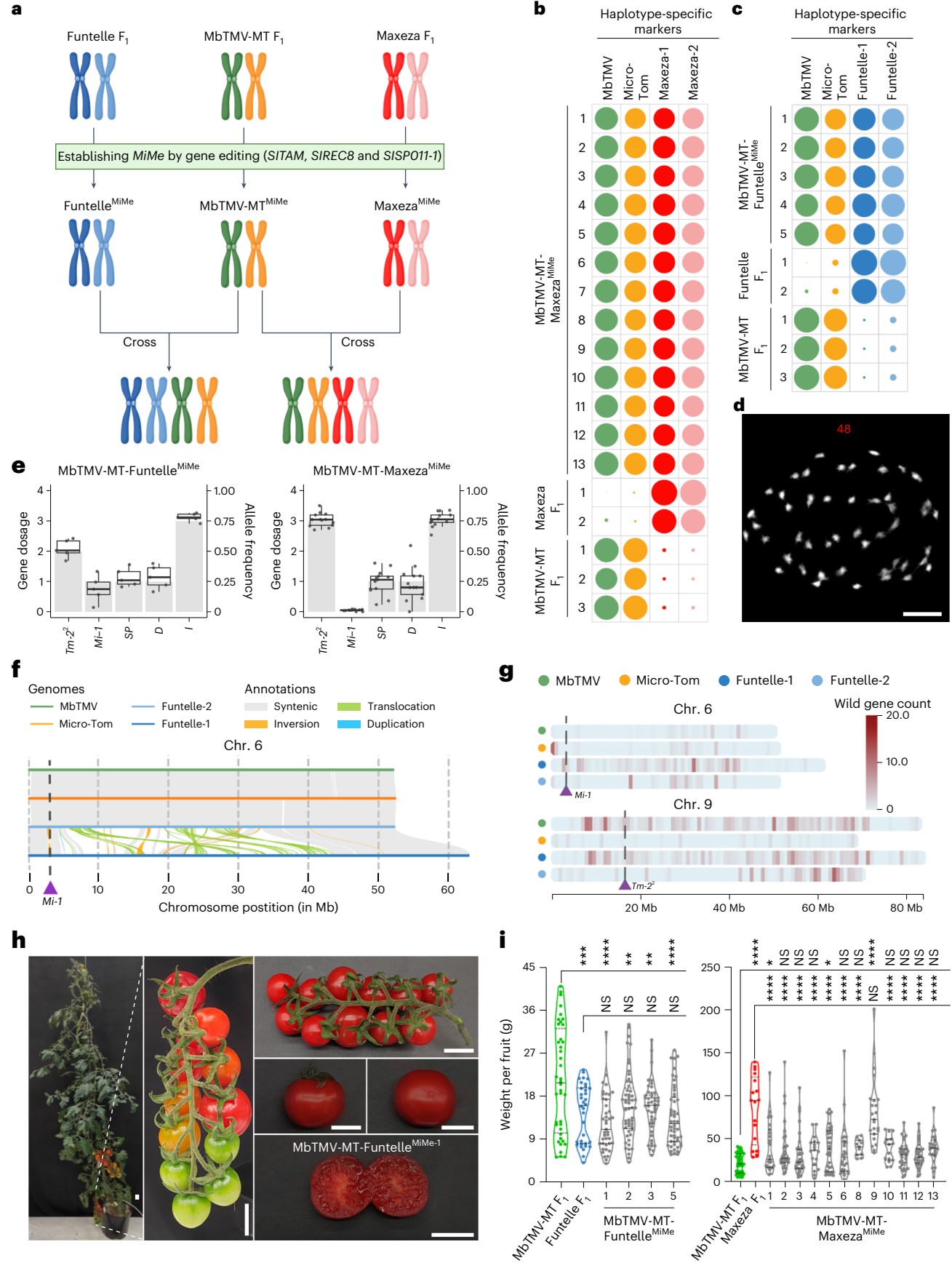

the clonal transfer of genomes from diploid wild materials into current polyploid crops (for example, strawberry) and the generation of highly heterozygous seedless triploid varieties (for example, banana). Hence, we propose that polyploid genome design can facilitate the controlled increase of genetic diversity in crops and open up completely novel breeding schemes.

## Online content

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

## Methods

### Plant growth and materials

Tomato plants were grown under long-day conditions (16 h light and 8 h dark) in Bronson chambers, Percival chambers and greenhouses at the Max Planck Institute for Plant Breeding Research. Tomato hybridization was accomplished by manual emasculation and pollination. For seed origin information, please refer to the Acknowledgements.

### Seed processing and germination

Seeds were collected from ripe tomato fruits and the pulp seed mixture was cleaned in a 1:1 (v/v) mix with 2% HCl solution for 30 min, followed by washing with fully desalinated water and drying overnight at room temperature. Thereafter, the harvested seeds were dried at room temperature for several weeks and stored at 4 °C. Seed germination was performed in vitro. Seeds were incubated in 1.5 ml sterile Milli-Q (Millipore) for 2 h. Subsequently, seeds were incubated in 1.5 ml saturated $Na_3PO_4$ buffer for 20 min, washed in Milli-Q three times and then incubated in 1.5 ml of 2.7% sodium hypochlorite (NaClO) for 20 min and washed three times using Milli-Q. Next, we transferred the seeds to 0.8% agar at 25 °C and then transferred the seedlings with roots into half-strength Murashige and Skoog (MS) solid medium in a culture room or soil in the greenhouse.

### Seed imaging

Seeds from wild-type Micro-Tom, *Sltam*, MbTMV-MT F$_1$ and MbTMV-MT$^{MiMe-A}$ mutants were imaged using a Leica M205 FA digital stereomicroscope (Leica Microsystems). Thereafter, Leica LAS X software was used to analyze the images. For quantitative seed size analysis, seed images were processed using the 'threshold' feature of ImageJ (https://imagej.net/software/fiji/downloads). Seed size area was measured using the 'analyze particles' feature, with a lower limit of '1-Infinity mm$^2$' to exclude any nonseed material. The final data were analyzed using Microsoft Excel and GraphPad Prism 9 software.

### CRISPR–Cas9 vector construction

The CRISPR–Cas9 system vector pDIRECT_22C[17] (containing a *35S::AtCAS9* cassette and in planta kanamycin resistance) was acquired from Addgene (plasmid no. 91135; https://www.addgene.org/91135/) and modified to knock out *SPO11-1*, *REC8*, *TAM* and *OSD1* (ref. 17). Specific single guide RNAs (sgRNAs) targeting each gene were designed using CRISPR-P v2.0 (http://crispr.hzau.edu.cn/CRISPR2/) and were selected to have a low rate of off-target mutagenesis in tomato[18]. The specificity of the designed sgRNAs was checked against various tomato genome assemblies, including BGV006775, BGV006865, BGV007931, BGV007989, Brandywine, Floradade, EA00371, EA00990, LYC1410, PAS014479, PI169588 and PI303721, using BLAT analysis[19]. The Golden Gate assembly approach was used to combine multiple sgRNAs into the destination expression vector[20]. The primers used for construct generation are listed in Supplementary Table 9 and the final constructs produced are listed in Supplementary Table 1.

### *Agrobacterium*-mediated transformation of tomato

CRISPR–Cas9 constructs were transformed into *Agrobacterium* strain GV3101 and incubated on YEP agar plates with antibiotic selection (10 µg ml$^{-1}$ gentamycin, 50 µg ml$^{-1}$ kanamycin and 20 µg ml$^{-1}$ rifampicin). PCR amplification was used to validate that transformed colonies contained plasmid. For tomato transformation, true leaves were taken from 4-week-old tomato plants grown in sterile culture and used as explants for transformation according to a previously described method[21]. Briefly, fresh leaves were cut into approximately 6 × 6 mm$^2$ pieces and incubated in MS-I medium (4.3 g l$^{-1}$ MS salt including vitamins, 100 mg l$^{-1}$ myo-inositol, 30 g l$^{-1}$ saccharose, 7 g l$^{-1}$ PhytoAgar, 2 mg ml$^{-1}$ zeatin riboside, 73 mg ml$^{-1}$ acetosyringon, 0.1 mg ml$^{-1}$ IAA, pH 5.9) overnight at 25 °C in the dark. *Agrobacterium* seed cultures were prepared by inoculation of liquid YEP with a single transformed colony, followed

by incubation overnight at 28 °C with shaking at 200 rpm. The next day, when the OD$_{600}$ reached between 0.4–0.6, liquid cultures were diluted 1:20 using liquid lysogeny broth (LB) without antibiotics. Dissected leaves were incubated in *Agrobacterium* solution for 15 min with occasional moderate shaking, followed by removal of excess liquid and incubation on MS-I medium for 48 h in the dark at 25 °C. Thereafter, leaf pieces were transferred onto MS-II (4.3 g l$^{-1}$ MS salt including vitamins, 100 mg l$^{-1}$ myo-inositol, 30 g l$^{-1}$ saccharose, 7 g l$^{-1}$ PhytoAgar, 1.5 mg ml$^{-1}$ zeatin riboside, 100 mg l$^{-1}$ kanamycin and 500 mg l$^{-1}$ carbenicillin, pH 5.9) plates and incubated under long-day conditions at 25 °C. The cultivation medium was refreshed every 2 weeks. Emerging shoots were transferred to MS-III rooting medium (2.15 g l$^{-1}$ MS salt including vitamins, 50 mg l$^{-1}$ myo-inositol, 15 g l$^{-1}$ saccharose, 7 g l$^{-1}$ PhytoAgar, 50 mg l$^{-1}$ kanamycin and 250 mg l$^{-1}$ carbenicillin, pH 5.9) and incubated under long-day conditions at 25 °C. Rooted plantlets were screened for transgenesis by PCR (*AtCAS9-F* and *AtCAS9-R*; *NPT-35S-F* and *NPT-35S-R*) and positive plants were transferred to soil.

### Alexander staining and scanning electron microscopy

Alexander staining was performed to determine pollen viability in individual tomato plants[22]. Mature pollen was collected from mature open flowers using a vibrating tool and stained using a commercial Alexander staining solution. Images were acquired using a Zeiss Axioplan 2 imaging fluorescence microscope equipped with a ZEISS Axiocam 208 color microscope camera, and the images were analyzed using Zeiss Labscope v3.1.

Scanning electron microscopy was performed as previously described[23]. Pollen grains of mature open flowers were collected into 2-ml Eppendorf tubes and then coated with palladium gold using a Polaron Sputter Coater SC 7600 (Quorum Technologies). The pollen was spread on 25.4-mm specimen mounts (or stubs) (Plano, no. G399F) using 25-mm conductive carbon adhesive tabs (Plano, no. G3348). The results were examined using a field emission scanning electron microscope (Supra 40 VP, Zeiss) with an acceleration voltage of 3 kV.

### High-throughput pollen size analysis

The Multisizer 4e (Beckman Coulter) was used to measure the particle diameter and volume of pollen samples derived from single open flowers[24]. Pollen was collected into 10 ml ISOTON II diluent (Beckman Coulter) in 25-ml Accuvette vials (Beckman Coulter) and measured according to the Multisizer 4e user's manual using a 100-µm aperture tube with an aperture current of 800 µA. Three different replications with an analytic volume of 1,500 µl were performed. Data were analyzed using Multisizer 4e and GraphPad Prism 9 software, and *P* values were calculated using standard one-way ANOVA.

### Statistics and reproducibility

For single-pollen diameter in wild-type and *MiMe* plants, we used the Wilcoxon rank-sum test in R (wilcox.test()). Exact *P* values cannot be reported because of the ties of data points within and between datasets. The following *P* values were calculated: wild type versus Micro-Tom$^{MiMe}$ ($<2 \times 10^{-16}$); MbTMV-MT F$_1$ versus MbTMV-MT$^{MiMe}$ ($<2 \times 10^{-16}$); Maxeza F$_1$ versus Maxeza$^{MiMe}$ ($<2 \times 10^{-16}$); Funtelle F$_1$ × Funtelle$^{MiMe}$ ($<2 \times 10^{-16}$).

For single-fruit weight data of 4-Hap plants, we used ordinary one-way ANOVA followed by Šídák's multiple-comparison test. For comparison of MbTMV-MT F$_1$ with MbTMV-MT-Funtelle$^{MiMe}$ genotypes, the following exact *P* values were calculated: MbTMV-MT F$_1$ versus Funtelle F$_1$ ($P = 0.004$); MbTMV-MT F$_1$ versus MbTMV-MT-Funtelle$^{MiMe-1}$ ($P = 5.54 \times 10^{-5}$); MbTMV-MT F$_1$ versus MbTMV-MT-Funtelle$^{MiMe-2}$ ($P = 0.0035$); MbTMV-MT F$_1$ versus MbTMV-MT-Funtelle$^{MiMe-3}$ ($P = 0.0012$); MbTMV-MT F$_1$ versus MbTMV-MT-Funtelle$^{MiMe-5}$ ($P = 2.79 \times 10^{-4}$). For comparison of Funtelle F$_1$ with MbTMV-MT-Funtelle$^{MiMe}$ genotypes, the following exact *P* values were calculated: Funtelle F$_1$ versus MbTMV-MT-Funtelle$^{MiMe-1}$ ($P = 0.65$); Funtelle F$_1$ versus MbTMV-MT-Funtelle$^{MiMe-2}$ ($P = 0.78$); Funtelle

$F_1$ versus MbTMV-MT-Funtelle[MiMe-3] ($P = 0.96$); Funtelle $F_1$ versus MbTMV-MT-Funtelle[MiMe-5] ($P = 0.96$). For comparison of MbTMV-MT $F_1$ with MbTMV-MT-Maxeza[MiMe] genotypes, the following exact $P$ values were calculated: MbTMV-MT $F_1$ versus Maxeza $F_1$ ($P = 3.25 \times 10^{-6}$); MbTMV-MT $F_1$ versus MbTMV-MT-Maxeza[MiMe-1] ($P = 0.02$); MbTMV-MT $F_1$ versus MbTMV-MT-Maxeza[MiMe-2] ($P = 0.05$); MbTMV-MT $F_1$ versus MbTMV-MT-Maxeza[MiMe-3] ($P = 0.69$); MbTMV-MT $F_1$ versus MbTMV-MT-Maxeza[MiMe-4] ($P = 0.44$); MbTMV-MT $F_1$ versus MbTMV-MT-Maxeza[MiMe-5] ($P = 0.02$); MbTMV-MT $F_1$ versus MbTMV-MT-Maxeza[MiMe-6] ($P = 0.25$); MbTMV-MT $F_1$ versus MbTMV-MT-Maxeza[MiMe-8] ($P = 0.24$); MbTMV-MT $F_1$ versus MbTMV-MT-Maxeza[MiMe-9] ($P = 2.39 \times 10^{-7}$); MbTMV-MT $F_1$ versus MbTMV-MT-Maxeza[MiMe-10] ($P = 0.18$); MbTMV-MT $F_1$ versus MbTMV-MT-Maxeza[MiMe-11] ($P = 0.69$); MbTMV-MT $F_1$ versus MbTMV-MT-Maxeza[MiMe-12] ($P = 0.65$); MbTMV-MT $F_1$ versus MbTMV-MT-Maxeza[MiMe-13] ($P = 0.05$). For comparison of Maxeza $F_1$ with MbTMV-MT-Maxeza[MiMe] genotypes, the following exact $P$ values were calculated: Maxeza $F_1$ versus MbTMV-MT-Maxeza[MiMe-1] ($P = 2.92 \times 10^{-4}$); Maxeza $F_1$ versus MbTMV-MT-Maxeza[MiMe-2] ($P = 1.35 \times 10^{-4}$); Maxeza versus MbTMV-MT-Maxeza[MiMe-3] ($P = 1.98 \times 10^{-5}$); Maxeza $F_1$ versus MbTMV-MT-Maxeza[MiMe-4] ($P = 6.01 \times 10^{-5}$); Maxeza $F_1$ versus MbTMV-MT-Maxeza[MiMe-5] ($P = 1.43 \times 10^{-4}$); Maxeza $F_1$ versus MbTMV-MT-Maxeza[MiMe-6] ($P = 7.69 \times 10^{-5}$); Maxeza $F_1$ versus MbTMV-MT-Maxeza[MiMe-8] ($P = 1.74 \times 10^{-4}$); Maxeza $F_1$ versus MbTMV-MT-Maxeza[MiMe-9] ($P = 0.74$); Maxeza $F_1$ versus MbTMV-MT-Maxeza[MiMe-10] ($P = 1.47 \times 10^{-4}$); Maxeza $F_1$ versus MbTMV-MT-Maxeza[MiMe-11] ($P = 2.14 \times 10^{-5}$); Maxeza $F_1$ versus MbTMV-MT-Maxeza[MiMe-12] ($P = 2.44 \times 10^{-5}$); Maxeza $F_1$ versus MbTMV-MT-Maxeza[MiMe-13] ($P = 1.34 \times 10^{-4}$). No fruit data were available for MbTMV-MT-Funtelle[MiMe-4] and MbTMV-MT-Maxeza[MiMe-7] because they did not grow to maturity.

### Ploidy determination and flow cytometry analysis
Ploidy in plants was determined using flow cytometry. In brief, one piece of fresh young tomato leaf was chopped using a sharp razor blade in 550 µl Galbraith's buffer (45 mM $MgCl_2$, 30 mM sodium citrate, 20 mM MOPS, 0.1% (v/v) Triton X-100, pH 7.0)[25]. Next, the slurry was passed through a 30-µm CellTrics green filter (04-0042, Sysmex). Subsequently, 20 µl DAPI (100 µg ml$^{-1}$) was added to 500 µl of the filtered sample, followed by incubation for 15 min. The CytoFLEX V5-B5-R3 flow cytometer was used following the manufacturer's instructions. Daily quality control was performed using CytoFLEX Daily QC fluorospheres (B53230). For data collection, 10,000 events per sample were acquired in fast mode for each independent measurement. The final data were analyzed using CytExpert (Beckman Coulter) and FCS Express 7 software.

### Chlorophyll content measurements
Leaf chlorophyll contents of control and 4-Hap plants were measured using the AtLEAF tool (https://www.atleaf.com/). atLEAF CHL values were converted into soil plant analysis development (SPAD) units using a web tool (https://www.atleaf.com/SPAD) and then converted into total chlorophyll content (mg cm$^{-2}$). For each plant, leaves were counted from the shoot apical meristem toward the ground to identify the fifth leaf (which was in all cases a mature leaf), which was used for analysis. Four different leaflets on each leaf were measured for chlorophyll content and six measurements were performed per leaflet (totaling 24 measurements per genotype).

### DNA extraction and sequencing of the CRISPR−Cas9 target sites
Genomic DNA was extracted using a BioSprint 96 DNA Plant Kit (Qiagen). Mutations at CRISPR−Cas9 target sites were analyzed using a combination of PCR amplification, gel electrophoresis and Sanger sequencing or Illumina sequencing. The presence of larger insertion/deletion mutations was first checked by staining agarose gels with Gelgreen (Sigma), imaging and analysis. Sanger sequencing was

conducted on PCR product diluted 1:30 using the Mix2Seq Kit (Eurofins Genomics). The produced ABI files were analyzed using the ICE analysis tool (Synthego; https://tools.synthego.com/#/). Illumina-based amplicon sequencing was modified from a previously described protocol[26]. M13F (TGTAAAACGACGGCCAGT) and M13R (CAGGAAACAGCTATGAC) were used as bridge sequences on target-specific PCRs and were later used to introduce Golay barcodes (from https://journals.asm.org/doi/10.1128/mSystems.00009-15) for each sample in a second round of PCR. After PCR purification, library construction was performed and Illumina sequencing (2 × 150-bp paired end) was done at the Max Planck Genome Centre in Cologne (MPGC Cologne). Sequencing data were analyzed using CLC Main Workbench 21.0.5 software (Qiagen) where the reads were demultiplexed and aligned to the reference (tomato genome, version SL4.0 and annotation ITAG4). In addition, fixed ploidy variant detection was performed to visualize the specific mutation sites of each gene among different chromosomes. The required variant probability parameter was more than 96% and the coverage and count filter was set to 5% minimum frequency.

### Introduction of tomato plants into in vitro culture and embryo rescue
Young side shoots of plants were selected and cut on a super-clean bench. Shoots were sterilized for 15 min in a 1:4 dilution of commercial bleach with 0.02% Tween and then washed three times with Milli-Q. This was followed by transplantation to 0.5 MS-10 medium and subculture on the same medium after 4−5 weeks.

### Chromosome spreads
Unopened flower buds (meiotic stages) were collected in 1.5 ml of Carnoy's fixation buffer (3:1 (v/v) absolute ethanol:acetic acid) and incubated under vacuum for 20 min. The buffer was then refreshed and the material was incubated overnight at room temperature until the tissue turned white. The fixation buffer was then replaced with 75% ethanol and samples were stored at 4 °C. Chromosome spreads were performed as described in a previous protocol[27,28] with the following modifications: the individual anthers were separated (the length of anthers ranges from 1.5 to 3 mm in meiotic stages) from the flower buds and digested with enzyme solution (0.3% (w/v) cellulase RS, 0.3% (w/v) pectolyase Y23, 0.3% (w/v) cytohelicase in citrate buffer, pH 4.7) for 2 h at 37 °C. Meiocytes were released from two or three anthers in 45% acetic acid by crushing the anthers using tweezers, followed by covering them with a coverslip, taking care to avoid bubbles. The slides were immersed in liquid nitrogen until no sound was heard and then the coverslip was removed. The slides were successively dried by applying 70%, 85% and 100% ethanol (5 min per concentration). Finally, 6 µl DAPI (10 µg ml$^{-1}$) was added to stain the chromosomes after the slides had dried. Images of chromosomal spreads were acquired using a Zeiss Axio Imager Z2 upright microscope and analyzed using ZEN blue software (Zeiss).

### Homologous protein sequence identification, characterization and phylogenetic tree analysis
The protein sequences of AtOSD1 (AT3G57860), AtCYCLIN A1;2 (AT1G77390), AtSPO11-1 (AT3G13170) and AtREC8/SYN1 (AT5G05490) were acquired from the *Arabidopsis* database TAIR (The *Arabidopsis* Information Resource; https://www.arabidopsis.org/) and then BLAST was performed against the phytozome database (https://phytozome-next.jgi.doe.gov/), the UniProt protein database (https://www.uniprot.org/blast), the Solanaceae Genomics Network database (https://solgenomics.net/) and the NCBI database (https://blast.ncbi.nlm.nih.gov/Blast.cgi) to identify homologous protein sequences in other species. Proteins were aligned using Clustal X2 followed by the construction of a phylogenetic tree using MEGA11. Gene structure images were created using the Exon-Intron Graphic Maker (http://wormweb.org/exonintron).

## Whole-genome sequencing of parental and offspring samples

High molecular weight DNA of Micro-Tom, Funtelle and Maxeza was isolated from 1.5 g of young leaf material using a NucleoBond HMW DNA kit (Macherey Nagel). DNA quality was assessed using a FEMTOpulse device (Agilent) and DNA quantity was measured using a Quantus Fluorometer (Promega). High-fidelity (HiFi) libraries were prepared according to the manual "Procedure & Checklist−Preparing HiFi SMRTbell Libraries using SMRTbell Express Template Prep Kit 2.0" with initial DNA fragmentation using a Megaruptor 3 (Diagenode) and final library size binning into defined fractions using SageELF (Sage Science). The size distribution was again controlled by FEMTOpulse (Agilent). Two size-selected libraries were sequenced per genotype on single SMRT cells (that is, a total of six SMRT cells) on a Pacific Biosciences Sequel II or Sequel IIe device at the MPGC Cologne with binding kit 2.0 and Sequel II Sequencing Kit 2.0 for 30 h (Pacific Biosciences).

For Micro-Tom, a chromatin conformation capture library was prepared using 0.5 g of young leaf material as the input. All treatments were performed according to the recommendations of the kit vendor (Omni-C, Dovetail) for plants. As a final step, Illumina-compatible libraries were prepared (Dovetail) and test-sequenced (2 × 150 bp paired end) on an Illumina NextSeq 2000 device at the MPGC Cologne, followed by sequencing on a NovaSeq 6000 at Novogene for increased coverage. In addition, Micro-Tom genomic DNA was extracted using the Qiagen DNeasy Plant Mini Kit and a BGI Plug-in Adapter Kit was used to prepare a sequencing library that was subsequently sequenced at BGI.

MbTMV, Funtelle and Maxeza genomic DNA was extracted using the DNeasy Plant Mini Kit (Qiagen). A PCR-free sequencing library was prepared at BGI and subsequently sequenced at BGI. MbTMV-MT $F_1$ (6 samples), MbTMV-MT $F_2$ (6 samples), MbTMV-MT$^{MiMe-A}$ offspring (6 samples), MbTMV-MT$^{MiMe-B}$ offspring (6 samples), Funtelle $F_1$ (2 samples), Maxeza $F_1$ (2 samples), MbTMV-MT$^{MiMe}$ × Maxeza$^{MiMe}$ offspring (13 samples) and MbTMV-MT$^{MiMe}$ × Funtelle$^{MiMe}$ offspring (5 samples) were profiled by Illumina sequencing. Briefly, small amounts (around 1 cm × 1 cm) of leaf samples were collected into 96-well plates and then the DNA was isolated at the MPGC Cologne using a NucleoMag Plant Kit (Macherey Nagel, 744400.4) to extract DNA on a robotic device (KingFisher, Thermo) followed by TPase-based DNA library preparation. The pooled libraries were sequenced using an Illumina NovaSeq 6000 machine at Novogene. Raw read and mapped read numbers are provided in Supplementary Table 7.

## Genome assemblies

For Micro-Tom, Hifiasm v0.16.1-r375 (ref. 29) was used with option -l0 to assemble the HiFi reads. First, omni-C (Dovetail) paired-end reads were mapped separately using Burrows−Wheeler Aligner v0.7 (ref. 30), followed by the addition of read mate scores, sorting and removal of duplicate reads using Samtools v1.9 (ref. 31). The resulting bam file was converted to a bed file and sorted (-k 4) using Bedtools v2.30 (ref. 32). Subsequently, one single round of Salsa v2.2 (ref. 33) was performed using the following optional settings: -e DNASE -m yes -p yes. A modified version of the convert.sh script was used to convert the Salsa2 output to a Hi-C file, which was used within a local installation of Juicebox (https://github.com/aidenlab/Juicebox) v1.11.08 to generate Hi-C contact plots. Assemblies scaffolded with the automated Salsa2 pipeline were further fine-tuned, with unplaced smaller scaffolds manually placed within larger scaffolds, based on Hi-C interaction and alignment to the MbTMV genome[9].

Haplotype-resolved assemblies of Funtelle and Maxeza $F_1$ hybrids were generated by running Hifiasm v0.16.1-r375 (ref. 29) on HiFi reads with default settings. Alignment of raw contigs against the MbTMV reference genome[9] was generated using minimap2 v2.24-r1122 (ref. 34) and visualized using D-GENIES v1.4.0. For each haplotype, contigs were anchored and scaffolded to chromosome-scale pseudomolecules using Ragtag v2.1.0 (ref. 35). $K$-mer analysis and genome size estimation were performed using genomescope v1.0 (ref. 36).

## Marker detection

We first identified and assigned SNP markers into two haplotypes. To detect markers between MbTMV and Micro-Tom, we first aligned the short reads (BGI) and long reads (HiFi) of both parental genomes against MbTMV[9] using bwa-mem v0.7.17 (ref. 37) and minimap2 v2.24-r1122 (ref. 29), respectively. SNPs were then detected using GATK HaplotypeCaller v4.2.4.1 and hard-filtering[38]. We retrieved homozygous Micro-Tom SNPs detected using both HiFi and Illumina reads that did not match any SNPs detected during MbTMV data alignment, ensuring that no ambiguous markers remained. We also selected markers that differed between the MbTMV-MT $F_1$ hybrid and the Funtelle $F_1$ hybrid by selecting homozygous Funtelle SNPs that did not overlap with any MbTMV or Micro-Tom SNPs. A similar approach was used to detect segregating markers between the MbTMV-MT $F_1$ hybrid and the Maxeza $F_1$ hybrid.

To identify unique markers in each of the four different haplotypes in the tetraploid tomato plants made by polyploid genome design (MbTMV-MT$^{MiMe}$ × Maxeza$^{MiMe}$ offspring; MbTMV-MT$^{MiMe}$ × Funtelle$^{MiMe}$ offspring), we initially needed to discover markers that differed between Funtelle haplotype 1 and Funtelle haplotype 2 and, separately, markers that differed between Maxeza haplotype 1 and Maxeza haplotype 2. In both cases, the same method was used, but for simplicity, the Funtelle case is described below. Both scaffolded assemblies of Funtelle haplotypes were aligned against MbTMV using minimap2 v2.24-r1122 and were then processed using MUMmer dnadiff[39] to identify SNPs. To confirm the SNPs were heterozygous between the two Funtelle haplotypes, we retrieved assembly-based SNPs that matched heterozygous SNPs reported from Funtelle HiFi and BGI reads. We retrieved those that did not overlap with any SNPs from MbTMV and Micro-Tom (Supplementary Fig. 25a). The resulting set of SNPs uniquely identified Funtelle haplotypes in both MbTMV and Micro-Tom. For unique Micro-Tom SNPs, we retrieved homozygous SNPs that did not overlap with any MbTMV or Funtelle SNPs. For unique MbTMV SNPs, we reported the genomic positions that showed homozygous SNPs for both Micro-Tom and Funtelle. To reduce false markers, we excluded a region on chromosome 9 between 5 Mb and 58 Mb, which is part of an introgression in MbTMV from the wild tomato relative *S. peruvianum*[9]. The resulting set of SNPs is considered to include unique parental markers.

## Determination of crossovers and aneuploidy

Trimming and quality checks of Illumina reads were done using Trim-Galore (https://github.com/FelixKrueger/TrimGalore). The alignment of reads and SNP calling against the MbTMV genome were performed using bwa-mem v0.7.17 and GATK HaplotypeCaller v4.2.4.1. For each sample, we computed the average allele frequency in a sliding 1-Mb window with a step size of 50 kb. We visually inspected deviations from the expected frequency to differentiate $F_2$ from $F_1$ and *MiMe* samples and to infer the presence or absence of recombination. Some of the samples showed unexpected allele frequencies in some genomic regions or chromosomes; therefore, we developed a script to identify regions with deviations in both allele frequency and average read coverage to infer chromosome fragmentation/aneuploidy. Finally, to determine whether all four haplotypes were present in the tetraploid tomato plants (MbTMV-MT$^{MiMe}$ × Maxeza$^{MiMe}$ offspring; MbTMV-MT$^{MiMe}$ × Funtelle$^{MiMe}$ offspring), we counted the number of unique parental markers that could be observed in each sample and compared this with the number in the control hybrids (MbTMV-MT $F_1$, Funtelle $F_1$ and Maxeza $F_1$).

## Predicting gene dosage and genotyping

We selected specific genes/regions of agronomic interest (tobacco mosaic virus resistance (*Tm-2*[2], Solyc09g018220)[40], *M. incognita* resistance (*Mi*, Solyc06g008720)[41,42], self-pruning (*SP*, Solyc06g074350)[43], dwarf (*D*, Solyc02g089160)[44] and *Fusarium* wilt resistance (*I*, Solyc11g011180)[19] to count alleles and dosage. For genes with known

causative mutations, we checked both the assemblies and BGI short reads. SyRI v1.6.3 was used to detect structural variations relative to the MbTMV reference and to find haplotypes with introgressions[45]. For the identified genomic rearrangements, genomic variations among the haplotypes, including syntenic regions, inversions, translocations and duplications, were highlighted in gray, yellow, light green and blue, respectively. SNP density was also computed to infer regions of wild introgression using ONT data from Fla.8111B and LA1589 to check for the presence of resistance gene $I$[19]. To directly genotype the same genes in each 4-Hap plant, we used either the known causative mutation or the haplotype of the resistant accession using GATK HaplotypeCaller v4.2.4.1. To annotate the assemblies and count the number of non-*Lycopersicum* genes, CDS sequences from *S. peruvianum*[46] were mapped to each of our parental assembles using minimap2. We identified wild-type genes within the large *S. peruvianum* introgressions on chromosomes 6 and 9.

## Reporting summary

Further information on research design is available in the Nature Portfolio Reporting Summary linked to this article.

## Data availability

Raw sequencing data of MbTMV and the MbTMV genome assembly are available at the European Nucleotide Archive (ENA) under project numbers PRJEB44956 and PRJEB63089. Raw sequencing data for Micro-Tom (PRJEB62441), Funtelle (PRJEB62442) and Maxeza (PRJEB62443) are available at ENA. Raw sequencing data for MbTMV-MT $F_1$ hybrids, MbTMV-MT $F_2$ offspring and all *MiMe* offspring (selfings and hybridizations) are available at the ENA under project number PRJEB63089. The genome assemblies of the hybrid varieties Funtelle and Maxeza (https://doi.org/10.5061/dryad.931zcrjs4)[47] and the genome assembly of inbred Micro-Tom (https://doi.org/10.5061/dryad.h9w0vt4qd)[48] are available at datadryad.org via the respective links.

## Code availability

All software used in the study is publicly available on the Internet, as described in the Methods section and Reporting Summary.

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

## Acknowledgements

MbTMV seeds were kindly provided for research purposes by C. Lelivelt and M. Verlaan (Rijk Zwaan, the Netherlands). Micro-Tom seeds were originally sourced from the Tomato Growers Supply Company (USA). Funtelle $F_1$ tomato seeds were kindly provided for research purposes by A. Voss (Syngenta, Germany). Maxeza $F_1$ tomato seeds were kindly provided for research purposes by M. van Stee (Enza Zaden, the Netherlands). We thank B. Huettel (MPGC Cologne) for next-generation sequencing library preparation and A. Stamatakis and the MPI-PZ greenhouse team for plant growth assistance. We appreciate the technical assistance of C. Sänger, O. Zabashta and M. Harperscheidt. We thank K. Choi and N. Donnelly for their comments on the manuscript. This study was funded by the Max Planck Society (Core funding to R.M. and C.J.U.), the German Research Foundation (DFG. grant UN 442 to C.J.U.) and the European Research Council (ERC starting grant 101076355, 'AsexualEmbryo', to C.J.U.). M.W.A.M.Z. was supported by a PhD fellowship from the Malaysian Agricultural Research and Development Institute and J.B.F. is supported by an Alexander von Humboldt fellowship.

## Author contributions

Y.W. identified target genes and generated the CRISPR–Cas9 constructs. Y.W. and S.E. performed plant transformations, plant regeneration, DNA extractions, seed collection and greenhouse plant management. Y.W. performed genotyping of CRISPR–Cas9 mutants and cytogenetic work. W.M.J.R. generated the Micro-Tom genome assembly and R.R.F. generated the Funtelle and Maxeza genome assemblies. R.R.F. performed marker identification, allele frequency analysis and gene dosage analysis. Y.W. and R.F. performed scanning electron microscopy. Y.W., S.E. and T.S. analyzed plant phenotypes. Y.W., R.R.F., M.W.A.M.Z., J.B.F., W.M.J.R., R.M. and C.J.U. performed data analysis and data interpretation. Y.W. and C.J.U. conceptualized the project and designed the experiments. C.J.U. directed the project. Y.W., R.R.F. and C.J.U. wrote the manuscript. All authors reviewed and approved the final manuscript.

## Funding

## Competing interests

C.J.U., Y.W. and R.M. are listed as inventors on European Patent Application no. 23179909.9, owned by the Max-Planck-Gesellschaft zur Förderung der Wissenschaften e.V., based on the results reported in this article. The other authors declare no competing interests.

## Additional information

**Extended data** is available for this paper at https://doi.org/10.1038/s41588-024-01750-6.

**Correspondence and requests for materials** should be addressed to Charles J. Underwood.

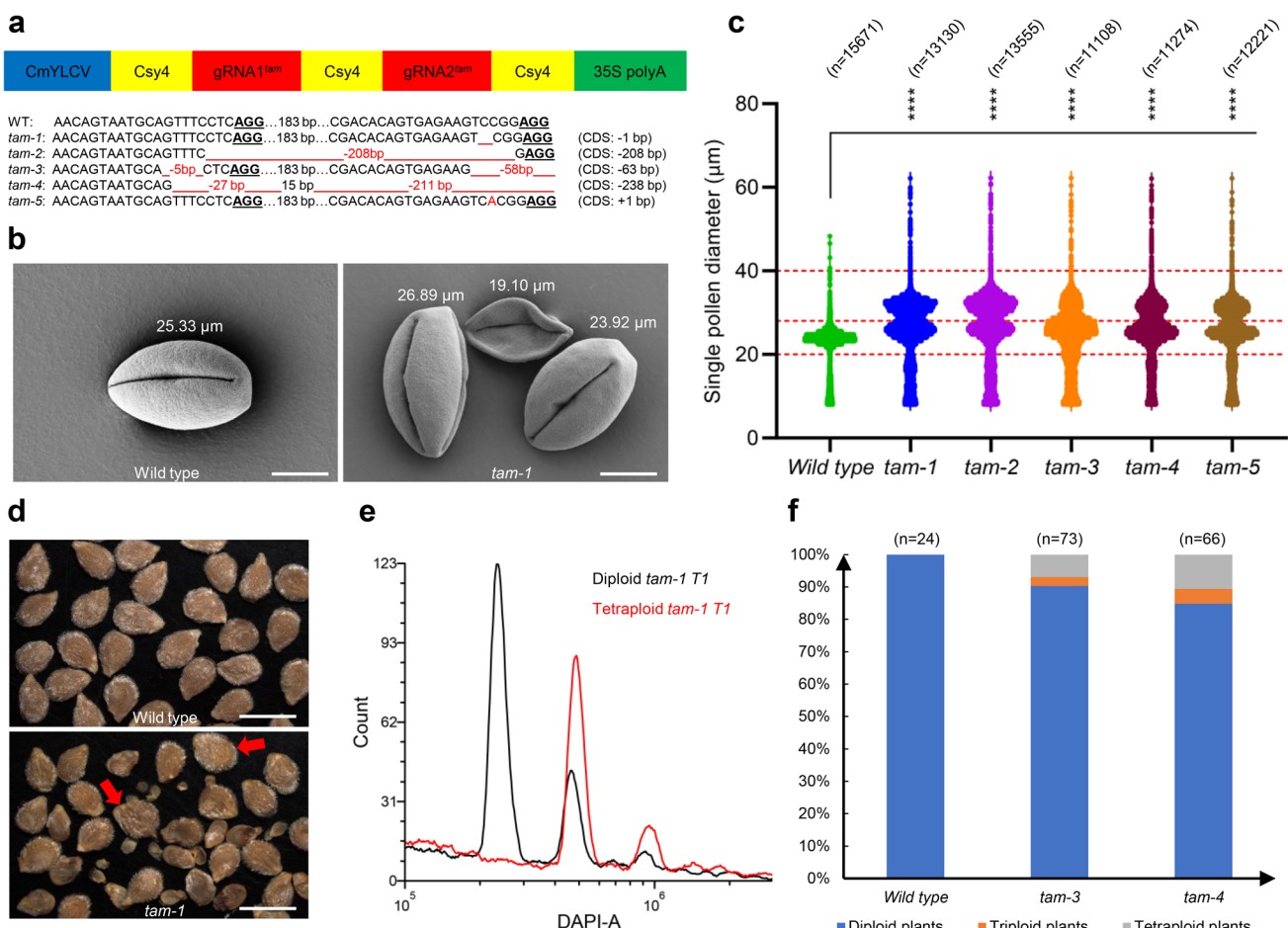

**Extended Data Fig. 1 | Unreduced gametes in *Sltam* mutants. a**, Top: Schematic representation of a CRISPR-Cas9 construct targeting the *SlTAM* gene in tomato. One Pol II promoter (CmYLCV) drives the expression of two sgRNAs (separated by a Csy4 spacer for gRNA production) that target the second exon of the *SlTAM* gene. Bottom: The sequence of wild type and five different alleles of *Sltam* are shown. The Protospacer Adjacent Motif (PAM) motif NGG for both gRNA targets is highlighted in bold and underlining. Mutation locations are indicated in red, with the total mutation length of the coding sequence (CDS) summarized at the end of the line. **b**, Scanning electron microscope images of wild type pollen (n = 49) and *tam-1* single mutant pollen (n = 56). Scale bars = 10 μm. **c**, Single pollen diameter distribution from single flowers collected from wild type and five independent alleles of *Sltam*. Reduced (haploid) pollen grains (20.03-28.15 μm) and unreduced (diploid) grains (28.15-40.23 μm) are highlighted by the red dotted lines. P values were calculated using Wilcoxon Rank-Sum test and **** means < 0.0001. No exact P values can be reported due to ties of data points within and between datasets. The following P values were calculated: wild type Vs *tam-1* (<2e-16); wild type Vs *tam-2* (<2e-16); wild type Vs *tam-3* (<2e-16); wild type Vs *tam-4* (<2e-16); wild type Vs *tam-5* (<2e-16). **d**, Image of seeds collected from self-fertilized (selfed) wild type (n = 28) and the *Sltam-1* mutant (n = 33). The red arrows indicate bigger seeds that give rise to tetraploid offspring. Scale bars = 3 mm. **e**, Flow cytometry of diploid (black) and tetraploid (red) selfed offspring of the *Sltam-1* T0 plant. Y axis of the histogram represents the events number and X axis of the histogram represents the intensity of DAPI signal. **f**, Ploidy level of selfing offspring from wild type Micro-Tom and two different *Sltam* alleles.

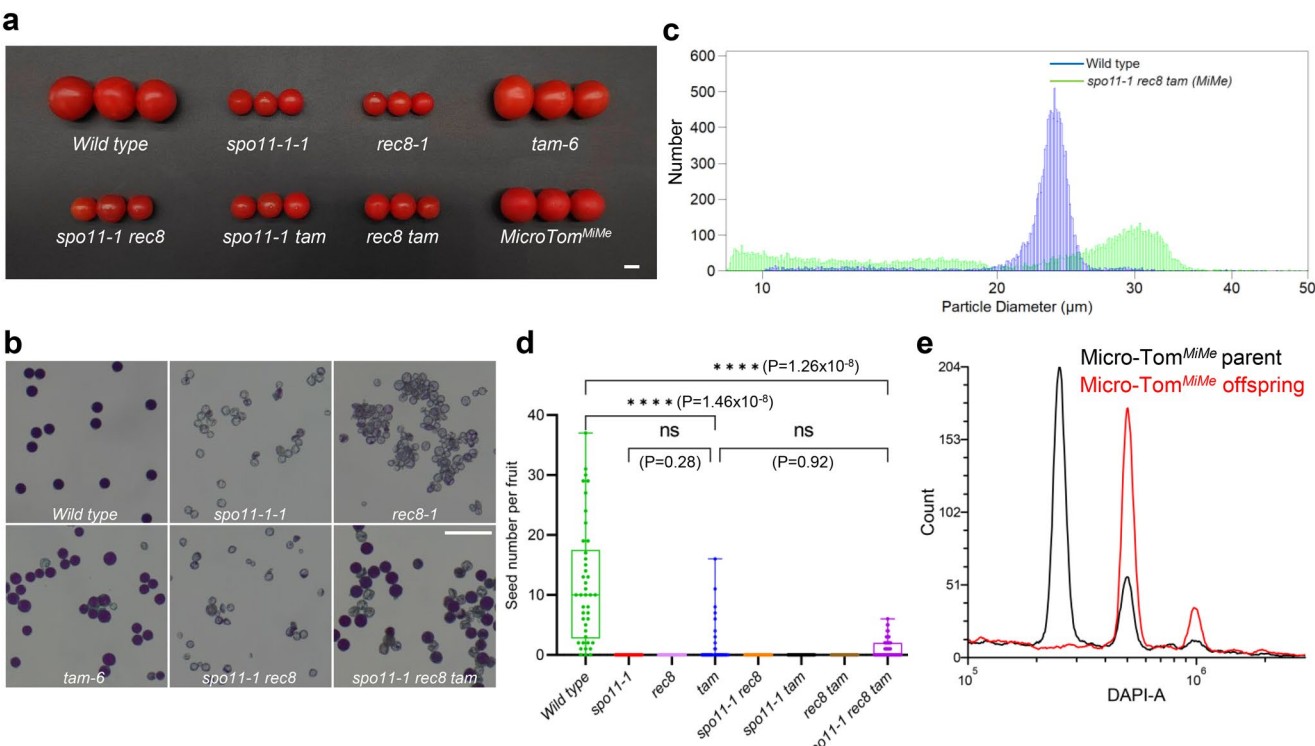

**Extended Data Fig. 2 | A *Mitosis instead of Meiosis* system in inbred tomato.**
**a**, Fruit shape overview of wild type, *spo11-1-1*, *rec8-1-1*, *tam-6*, *spo11-1 rec8*, *spo11-1 tam*, *rec8 tam* and *spo11-1 rec8 tam* mutants. Scale bar = 1 cm. **b**, Alexander staining results of wild type, single *spo11-1-1* mutants, single *rec8-1* mutants, single *tam-6* mutants, double *spo11-1 rec8* mutants and triple *spo11-1 rec8 tam* mutants' pollen. Scale bar = 200 μm. **c**, Single pollen diameter distribution from single flowers of wild type (n = 12191) and triple *spo11-1 rec8 tam* (inbred *MiMe*) mutant (n = 12149). **d**, Seed number per single fruit of wild type (n = 42, 11.74 ± 1.54 SEM), *spo11-1* (n = 67, 0 ± 0 SEM), *rec8* (n = 89, 0 ± 0 SEM), *tam* (n = 67, 0.88 ± 0.34 SEM),

*spo11-1 rec8* (n = 85, 0 ± 0 SEM), *spo11-1 tam* (n = 41, 0 ± 0 SEM), *rec8 tam* (n = 43, 0 ± 0 SEM) and *spo11-1 rec8 tam* (n = 153, 0.84 ± 0.11 SEM). Each dot indicates the seed number of an individual fruit, the solid line represents the median, boxes show quartiles and whiskers show maximum and minimum values. P values are from Ordinary one-way ANOVA followed by Šídák's multiple comparisons test. 'ns' means no significance, and **** indicates a P value < 0.0001. **e**, Flow cytometry analysis of diploid parent Micro-Tom[MiMe] (black) and tetraploid Micro-Tom[MiMe] offspring (red). Y axis of the histogram represents the events number and X axis of the histogram represents the intensity of DAPI.

# Reporting Summary

## Statistics

For all statistical analyses, confirm that the following items are present in the figure legend, table legend, main text, or Methods section.

| n/a | Confirmed | |
|---|---|---|
| ☐ | ☒ | The exact sample size (*n*) for each experimental group/condition, given as a discrete number and unit of measurement |
| ☐ | ☒ | A statement on whether measurements were taken from distinct samples or whether the same sample was measured repeatedly |
| ☐ | ☒ | The statistical test(s) used AND whether they are one- or two-sided *Only common tests should be described solely by name; describe more complex techniques in the Methods section.* |
| ☐ | ☒ | A description of all covariates tested |
| ☐ | ☒ | A description of any assumptions or corrections, such as tests of normality and adjustment for multiple comparisons |
| ☐ | ☒ | A full description of the statistical parameters including central tendency (e.g. means) or other basic estimates (e.g. regression coefficient) AND variation (e.g. standard deviation) or associated estimates of uncertainty (e.g. confidence intervals) |
| ☐ | ☒ | For null hypothesis testing, the test statistic (e.g. *F*, *t*, *r*) with confidence intervals, effect sizes, degrees of freedom and *P* value noted *Give P values as exact values whenever suitable.* |
| ☒ | ☐ | For Bayesian analysis, information on the choice of priors and Markov chain Monte Carlo settings |
| ☒ | ☐ | For hierarchical and complex designs, identification of the appropriate level for tests and full reporting of outcomes |
| ☒ | ☐ | Estimates of effect sizes (e.g. Cohen's *d*, Pearson's *r*), indicating how they were calculated |

*Our web collection on statistics for biologists contains articles on many of the points above.*

## Software and code

Policy information about availability of computer code

| Data collection | High throughput data of single pollen diameter was collected via a Multisizer 4e (Beckman Counter, Germany). Scanning electron microscopy (SEM) data was obtained by emission scanning electron microscope. Images of spread meiotic chromosomes were captured using a Zeiss Axio Imager Z2 upright microscope. Ploidy determination of plants was carried out using flow cytometry of leaf nuclei via DAPI staining. DNA sequencing was carried out on Illumina, MGI and PacBio HiFi platforms. Seeds from wild type and mutants were imaged using a Leica M205 FA digital stereomicroscope (Leica Microsystems, Germany). Leaf chlorophyll contents of control and 4-Hap plants were measured via a powerful tool AtLEAF (https://www.atleaf.com/, US). |
|---|---|

| Data analysis | Fruit, seed and pollen data analysis: Graphpad Prism 9, Leica Application Suite X v3.7.3.23245 (LAS X), ImageJ 1.51u (Fiji), Zeiss Labscope v3.1; Phylogenetic tree: ClustalX2, MEGA11; Chromosome data: ZEN 3.5 (blue edition); Single pollen size: Multisizer 4e v4.04, Graphpad Prism 9; Illumina sequencing data: CLC Main Workbench 21.0.5; Ploidy determination and flow cytometry analysis: FCS Express 7, CytExpert v2.4.0.28 (Beckmann Counter, Germany) Mutiple protein sequence analysis: ClustalX2, BioEdit v7.2.0; Genome assemblies and downstream analysis: Hifiasm v0.16.1-r375, Salsa v2.2, Juicebox v1.11.08,  Burrows-Wheeler Aligner v0.7, Samtools v1.9, Bedtools v2.30, minimap2 v2.24-r1122, D-GENIES v1.4.0, RagTag  v2.1.0, genomescope v1.0; Marker detection: bwa-mem v0.7.17 and minimap2 v2.24-r1122; Determination of crossovers: GATK HaplotypeCaller v4.2.4.1; Detection of genomics structural variations: SyRI v1.6.3. |
|---|---|

For manuscripts utilizing custom algorithms or software that are central to the research but not yet described in published literature, software must be made available to editors and reviewers. We strongly encourage code deposition in a community repository (e.g. GitHub). See the Nature Portfolio guidelines for submitting code & software for further information.

## Data

Policy information about availability of data

All manuscripts must include a data availability statement. This statement should provide the following information, where applicable:

- Accession codes, unique identifiers, or web links for publicly available datasets
- A description of any restrictions on data availability
- For clinical datasets or third party data, please ensure that the statement adheres to our policy

Raw sequencing data of MbTMV and the MbTMV genome assembly are available at the European Nucleotide Archive (ENA) under project numbers PRJEB44956 and PRJEB63089. Raw sequencing data of Micro-Tom (PRJEB62441), Funtelle (PRJEB62442) and Maxeza (PRJEB62443) are available at the ENA. Raw sequencing data of MbTMV-MT F1 hybrids, MbTMV-MT F2 offspring and all MiMe offspring (selfings and hybridizations) are available at the ENA under project number PRJEB63089.
Dryad Submission entitled "PacBio HiFi based haplotype-aware assemblies of tomato hybrid varieties Funtelle and Maxeza" with a unique digital object identifier (DOI): https://doi.org/10.5061/dryad.931zcrjs4
Dryad Submission entitled "A chromosome-scale de novo genome assembly of the dwarf tomato variety Micro-Tom" with a unique digital object identifier (DOI): https://doi.org/10.5061/dryad.h9w0vt4qd
The protein sequences were acquired from the Arabidopsis database TAIR (The Arabidopsis Information Resource, https://www.arabidopsis.org/) and then protein BLAST was performed against the phytozome database (https://phytozome-next.jgi.doe.gov/), the UniProt protein database (https://www.uniprot.org/blast), the Solanaceae Genomics Network database (https://solgenomics.net/) and the NCBI database (https://blast.ncbi.nlm.nih.gov/Blast.cgi) to identify homologous protein sequences in other species. Protein sequences alignment were achieved using Clustal X2 followed by construction of phylogenetic tree using MEGA11. Gene structure images were created using Exon-Intron Graphic Maker (http://wormweb.org/exonintron).

## Research involving human participants, their data, or biological material

Policy information about studies with human participants or human data. See also policy information about sex, gender (identity/presentation), and sexual orientation and race, ethnicity and racism.

| Reporting on sex and gender | NA |
|---|---|
| Reporting on race, ethnicity, or other socially relevant groupings | NA |
| Population characteristics | NA |
| Recruitment | NA |
| Ethics oversight | NA |

Note that full information on the approval of the study protocol must also be provided in the manuscript.

## Field-specific reporting

Please select the one below that is the best fit for your research. If you are not sure, read the appropriate sections before making your selection.

☒ Life sciences          ☐ Behavioural & social sciences          ☐ Ecological, evolutionary & environmental sciences

For a reference copy of the document with all sections, see nature.com/documents/nr-reporting-summary-flat.pdf

# Life sciences study design

All studies must disclose on these points even when the disclosure is negative.

| | |
|---|---|
| Sample size | We made use of one model hybrid (Moneyberg-TMV x Micro-Tom) and two commercial F1 hybrid lines (Funtelle and Maxeza) in this study without making a power calculation. By generating MiMe triple mutants in these three genetic backgrounds that represent highly divergent tomato genotypes we could demonstrate the MiMe phenotype was robust to different genetic backgrounds. The exact sample size for each experiment (single pollen particle size measurement, flow cytometry, embryo rescue, cytology) was clearly mentioned in the main text, figure legend or methods. |
| Data exclusions | No data were excluded. |
| Replication | Replications for each experiment were clearly stated in main text, figure legends or Methods section. For tam mutants, we tested five different alleles for experiments. At least three independent samples (biological replicates) and three replicate samples (technical replicates) were performed for all experiments. |
| Randomization | For the phenotype testing of F1 hybrid, F2 offspring and hybrid MiMe offspring, location of plants in the greenhouse was random. Meanwhile, all experiments plants were grown in consistent conditions, with appropriate controls grown side-by-side in Bronson Chamber, Percival Chamber and greenhouse. |
| Blinding | Where relevant blinding was carried out including during fruit weight analysis and other plant phenotyping experiments during data collection by technicians. The watering and plant nutrient irrigation system was performed without knowledge of plant genotype by greenhouse gardeners. In any non-blinded analysis several authors reviewed the data to ensure robust data analysis had been carried out. |

# Reporting for specific materials, systems and methods

We require information from authors about some types of materials, experimental systems and methods used in many studies. Here, indicate whether each material, system or method listed is relevant to your study. If you are not sure if a list item applies to your research, read the appropriate section before selecting a response.

## Materials & experimental systems

| n/a | Involved in the study |
|---|---|
| ☒ | ☐ Antibodies |
| ☒ | ☐ Eukaryotic cell lines |
| ☒ | ☐ Palaeontology and archaeology |
| ☒ | ☐ Animals and other organisms |
| ☒ | ☐ Clinical data |
| ☒ | ☐ Dual use research of concern |
| ☐ | ☒ Plants |

## Methods

| n/a | Involved in the study |
|---|---|
| ☒ | ☐ ChIP-seq |
| ☐ | ☒ Flow cytometry |
| ☒ | ☐ MRI-based neuroimaging |

## Flow Cytometry

### Plots

Confirm that:

☒ The axis labels state the marker and fluorochrome used (e.g. CD4-FITC).

☒ The axis scales are clearly visible. Include numbers along axes only for bottom left plot of group (a 'group' is an analysis of identical markers).

☐ All plots are contour plots with outliers or pseudocolor plots.

☐ A numerical value for number of cells or percentage (with statistics) is provided.

### Methodology

| | |
|---|---|
| Sample preparation | One piece of fresh young tomato leaf (2cm x 3mm) was chopped using a sharp razor blade in 550 μL Galbraith's buffer (45 mM MgCl2, 30 mM sodium citrate, 20 mM MOPS, 0.1% (v/v) Triton X-100, pH7.0) (Galbraith, D. W. et al. 1983). Next the slurry was passed through a 30-μm CellTrics green filter (REF: 04-0042-2316, Sysmex). Subsequently, 20 μL DAPI (100 μg/mL) was added to 500 μL filtered sample, followed by incubation for 15 minutes and run on the CytoFLEX V5-B5-R3 flow cytometer following manufacturer's instructions. |
| Instrument | CytoFLEX V5-B5-R3 flow cytometer |
| Software | CytExpert, FCS Express 7 |
| Cell population abundance | The plant ploidy levels were determined from tomato leaf nuclei. In this ploidy checking experiment, cell sorting and |

| Cell population abundance | purification steps were not carried out. After stable peaks were formed, 10,000 events per sample were acquired in fast mode for each independent measurement. |
| --- | --- |
| Gating strategy | Gating was used to ensure that non-nuclear particles/debris with weak DAPI staining were not considered as plant nuclei. An example of the gating strategy used is presented in the Supplementary Figure 6. |

☒ Tick this box to confirm that a figure exemplifying the gating strategy is provided in the Supplementary Information.

