## [Peer Review File · Nature Genetics]

Peer Review Information

Manuscript Title: Harnessing clonal gametes in hybrid crops to engineer polyploid genomes

Corresponding author name(s): Dr Charles (J.) Underwood

Reviewer Comments & Decisions:

Decision Letter, initial version:

30th Aug 2023

Dear Dr Underwood,

Your Brief Communication, "Harnessing clonal gametes in hybrid crops to engineer polyploid genomes" has now been seen by 4 referees. You will see from their comments below that while they find your work of interest, some important points are raised. We are interested in the possibility of publishing your study in Nature Genetics, but would like to consider your response to these concerns in the form of a revised manuscript before we make a final decision on publication.

To guide the scope of the revisions, the editors discuss the referee reports in detail within the team with a view to identifying key priorities that should be addressed in revision. In this case, we think all four referees have provided constructive reviews aimed at strengthening the analyses and improving the presentation, and we particularly ask that you address their technical comments as thoroughly as possible with appropriate revisions. We hope that you will find the prioritized set of referee points to be useful when revising your study.

We therefore invite you to revise your manuscript taking into account all reviewer and editor comments. Please highlight all changes in the manuscript text file. At this stage we will need you to upload a copy of the manuscript in MS Word .docx or similar editable format.

*2) If you have not done so already please begin to revise your manuscript so that it conforms to our Brief Communication format instructions, available here.

*3) Include a revised version of any required Reporting Summary:

Please be aware of our guidelines on digital image standards.

[redacted]

We hope to receive your revised manuscript within 3 to 6 months. If you cannot send it within this time, please let us know.

Sincerely,
Wei

Wei Li, PhD
Senior Editor
Nature Genetics
New York, NY 10004, USA
www.nature.com/ng

Reviewers' Comments:

Reviewer #1:

Remarks to the Author:

The authors describe a system for APH (autopolyploid progressive heterosis) in tomato, by sequentially combining 4 different haplotypes into a tetraploid plant. They utilize the MiMe approach that was previously demonstrated to produce unreduced gametes in Arabidopsis and rice. The importance of this system is that it will allow for combinations of a much greater diversity of alleles (including from wild relatives) and more heterotic effects than is possible with conventional breeding.

Their previous MiMe design required knockout of the OSD1 gene, but the authors determined that this was not possible in tomato, because it is a single copy essential gene in tomato (unlike Arabidopsis and rice). They then show that disruption of the tomato TAM/Cyclin A1 (also a single copy gene) leads to diploid gametes, similar to OSD1. In combination with knockouts of REC8 and SPO11, they get clonal unreduced gametes, with significant pollen lethality but still producing viable seeds that are tetraploid. They then use this approach to make MiMe triple knockouts in three different F1 hybrid genotypes, including 2 commercial hybrids. The MiMe hybrids makes clonal unreduced gametes, and when cross-pollinated gives rise to tetraploid progeny with unrecombined haplotypes from all 4 parents with high frequency. Plants with the intact genomes of all four haplotypes were obtained and confirmed by whole genome sequencing, that showed normal vegetative growth and produced seedless fruit.

Overall, this study – although submitted as a Brief Communication - is impressively thorough (with 29 Supplementary figures!), and the major results are clearly presented and convincing. Although the system will need improvement before it can be used in actual cultivation, it provides important proof of concept for an approach that opens the door to new breeding schemes that can exploit allelic diversity and heterosis at the polyploid level. Their scheme is potentially applicable to other dicot crops, and as the authors point out, it can be especially useful for crops like potato that can be vegetatively propagated after constructing favourable combinations of parental genomes.

Comments:

1. The analysis of mutant plants is focused on the male gametes. A major omission is that there is no information about their effects on the female gametes. It is understandable that detailed characterization of female meiosis and gametes was not undertaken, because it is much more difficult. But indirect inferences can be drawn from their own data. The occurrence of tetraploid progeny implies that some egg cells had to be diploid. This should be stated explicitly, and estimates of the fractions of unreduced clonal female gametes for their different TAM and MiMe mutants should be provided, which is important for others who are considering use of this system.
2. To follow up on the previous comment, Extended Data Fig. 1 shows tetraploid frequencies of 10-15% for two TAM mutants, whereas main Fig. 1h shows 93% tetraploid frequency for a MiMe mutant. Why would the tetraploid frequency increase? Is it because mutations in SPO11 and REC8 are lethal unless rescued by loss of TAM function? If so, this would account for the very low seed set, because low penetrance of the TAM mutation during female meiosis would lead to mostly dead seeds, the only survivors would be tetraploid.

3. Since the authors have extensively used pollen size as a convenient proxy for ploidy, they should provide a reference where correlation between size and ploidy was established (preferably for tomato pollen). Can the possibility that some larger pollen grains are aneuploid be excluded?
4. The tam-6 allele appears in multiple figures, but its origin is buried in Supp. Table 2. It should be mentioned up front, in the Supplementary Note.
5. Fig. 1c shows that MiMe in the commercial hybrids results in high pollen lethality, which is not observed with MiMe in MT. Can the authors comment on this difference? E.g. Is the tam mutation less penetrant in the hybrids?
6. Page 2 lines 12-13: "MbTMV-MT MiMe mutants produced smaller fruits that contained less seeds (Fig. 1 e,f)". Should say "much fewer seeds" based on Fig. 1f.
7. It would be helpful if the authors could discuss why deletions and truncations sometimes arise in their MiMe plants, and to what extent these are likely to pose a problem when scaling up for practical applications.
8. For the analysis of read coverage/SNP changes, they have used a 1 Mb sliding window, which may not give enough resolution to find smaller deletions or truncations. The authors should mention this issue in the Supplementary Note.
9. Suppl. Fig. 20 could use a clearer and more explanatory legend summarizing the conclusions from the data presented here. Also, the colours are difficult to distinguish, especially in 20b. The other figures similar to this one would also benefit from more explanatory legends, as would Suppl. Figures 22-23 (e.g., what do they conclude from these figures).
10. Suppl. Fig. 29: Though it is mentioned in the Methods section, the figure legend should additionally include the significance of the Tm-2 introgression, for the benefit of researchers not familiar with tomato genetics.
11. Page 3 line 24 – What is meant by "harness heterosis – on farm – at the polyploid level"? Do they mean "harness heterosis at the polyploid level" for farming / agriculture?

Reviewer #2:

Remarks to the Author:

In this manuscript by Wang and colleagues, the authors set out to apply the "Mitosis instead of Meiosis" (MiMe) system that has been previously published in Arabidopsis and the monocot crop rice to the dicot crop tomato. To establish the MiMe system in tomato, the authors first identify homologs of AtOSD1, AtREC8, and AtSPO11-1, the genes which have been mutated for MiMe in Arabidopsis. The authors find that mutants lacking SIOSD1 activity are lethal, and explore another mutation that has been previously shown to skip the meiotic division in Arabidopsis (tam1). The authors then demonstrate in an elegant way that the MiMe system with SISPO11-1, SIREC8, and SITAM allows the production of unreduced clonal gametes in tomato and the "fixation" of hybrid genotypes. Furthermore, the authors cross distinct MiMe hybrid genotypes and produce viable tetraploid offspring that combines haplotypes from 4 distinct parents.

The application of the MiMe system in a dicot crop and the combination of agronomically-important alleles from 4 distinct haplotypes by hybridization of clonal gametes from two hybrids is novel. However, whether heterosis effects can be conserved through the MiMe approach remains unclear. The authors use state-of-the-art technologies to produce high-quality data, which leads them to mainly sound conclusions. In parts, the manuscript lacks clarity, mainly because of its density and the organization of the Suppl. Data. I have several comments about the presentation and interpretation of the data, which I outline in detail below.

Major comments

page 1, line 43-46: The authors report that they established the MiMe system in tomato by mutating the three genes SISPO11-1, SIREC8, and SITAM. They refer in a single sentence to 1 Supplementary Note, 3 panels of Fig. 1, 2 Extended Data Figures, and 15 Suppl. Figures. While their work on SIOSD1 and SITAM is described in more detail in the Suppl. Note, their data and findings on SISPO1 and SIREC8 remains largely un-commented. The authors should describe the data in Suppl. Fig. 10-15 and their conclusions either in the Main Text or Suppl. Note. I also suggest to include the two Extended Data Figures to the list of Supplementary Figures to make the manuscript easier accessible to the reader.

page 2, line 3: The author write that they “identified six MbTMV-MT, three Funtelle and three Maceza lines that had biallelic 4 mutations on SISPO11-1, SIREC8, and SITAM (Supplementary Table 4).” It is unclear from the main text and methods how the mutations were induced (multiplex, number of gRNAs, etc.). In addition, Suppl. Table 4 shows only data for 4 MbTMV-MT lines, one Funtelle, and one Maceza line. The MbTMV-MT lines A, C, and D show identical alleles for all three genes suggesting that they descend from a single transformation event and are not independent events. The authors should clarify.

page 2, line 13-14: The authors report that the MbTMV-MT MiMe plants produced smaller fruits with larger seeds. Quantitative data for the seed size increase should be presented to support the claim.

page 2, line 23-24: The authors write that they “validated the partial loss of one of the MbTMV copies of chromosome 9 (Fig. 1i).” However, the panel of Fig.1i rather shows the observation of the loss based on allele frequency data. The authors should either change their wording or validate the partial loss using an alternative approach.

page 2, line 38: The authors write that they generated a platinum-grade assembly for Micro-Tom. Assembly statistics should be included in the manuscript to allow the reader to assess the quality of the assembly.

Fig. 2f: The Figure shows major structural variation on chromosome 6 between the two Funtelle haplotypes. However, it is unclear how the data supports the presence of Mi resistance variant. Please explain.

page 3, line 6-7 and Fig. 2g: It is unclear how the authors conclude from Fig. 2g that “MbTMV-MT-MaxezaMiMe plants have more Tomato Mosaic Virus (TMV) resistance alleles than MbTMV-MT-FuntelleMiMe” or whether Fig. 2g supports another conclusion.

Minor comments:

page 1, line 46: The authors explain that they implemented the MiMe system in three agronomically-

relevant hybrid genotypes. One of those is a Moneyberg-TMV x Micro-Tom hybrid. Can the authors explain the agronomic relevance of this hybrid given that Micro-Tom is mainly a model for basic research?

page 2, line 12. The authors find that the MbTMV-MT MiMe plants develop smaller fruits than the MbTMV-MT parents. Can the authors speculate why fruits are smaller and how this negative effect could be overcome?

page 3, line 9. The authors write that the 4-H plants developed "well-organized branches with seedless fruits". The authors should specify whether they refer to reproductive branches (inflorescences) or vegetative branches (shoots). In addition, the authors could include images of the parental genotypes in Fig. 2 and Suppl. Fig. 25 as comparison to allow the reader to determine whether development of shoots, inflorescences, and fruits are abnormal in the 4-H plants.

page 13, line 1: The authors should make all assemblies that were newly generated in this study available to the community.

Figure 1a: The schematics of tomato plants suggest that tomato plants develop fruits and flowers on their compound leaves, which could lead to botanical misconceptions.

Reviewer #3:

Remarks to the Author:

This study uses the MiMe system of bypassing meiosis to produce tetraploids with four distinct genomes that would avoid the problems of meiotic disturbances in autotetraploids and that would be uniform. Producing a double cross tetraploid by sexual means will introduce variability in a quadruplex hybrid due to double reduction in the hybrid tetraploid parents and aneuploid progeny from segregations that are not 2:2 because of different types of associations of the four homologues in meiosis. The authors have achieved their goal. Some thoughts on the presentation follow.

The manuscript is very densely written and could probably be expanded with greater explanations that would be helpful to the more general reader. It is classified as a Brief Communication but perhaps the editors could allow an expansion.

The authors use the term 4-H to designate a tetraploid with four distinct genomes. The classical term for this configuration is a quadruplex tetraploid. It is fine to use the 4-H designation, but the authors should be aware of the priority term. (The authors should also be aware that "4-H" is a youth organization in the United States.)

In Figure 1j, it is difficult to distinguish between the F1 and F2 lines. The green and blue lines appear very similar. The authors might consider a more distinctive color difference.

In the Introduction, the authors note that the system could help capitalize on progressive heterosis and this reviewer agrees that this procedure would be very beneficial for making it more uniform. But it was not clear whether there was progressive heterosis in the 4-H material produced. In Figure 2h there is no comparison to 2-H tetraploids and the data in Figure 2i do not make it clear. Perhaps there is midparent heterosis? At any rate, by including MicroTom, the biomass characters might be too

skewed for progressive heterosis to be obvious. When the four grandparents are more comparable, then it is more obvious.

On page 3 lines 26-27: cultivated strawberry is an allopolyploid (Nature Genetics, 2019, 51: 541-547). The seedless nature of banana has as much to do with parthenocarpy as to its triploid nature. Some bananas have three genomes of *Musa acuminata* while others are triploid hybrids of *M. acuminata* and *M. balbisiana* (See references in D'Hont et al 2012, Nature 488: 213-217). The sentence might be taken to suggest that they are all autopolyploids.

Reviewer #4:

Remarks to the Author:

MiMe (Mitosis instead of Meiosis) established in rice and Arabidopsis has provided the chance of transmission of hybrid genome into unreduced gametes without recombination. In addition to MiMe, parthenogenetic development of unreduced egg cell triggered by ectopic expression of an AP2-type transcription, such as *OsBBML1*, is required for formation of clonal rice seeds (progenies). In the present manuscript (NG-BC62972R), the authors introduced MiMe into tomato to establish the artificial building-up system of autopolyploid progressive heterosis (APH). Although MiMe introduction into tomato is not new/unique concept, the authors successfully combined haplotypes of interests without recombination in tomato. These MiMe based APH in tomato may provide a horizon for utilization of haplotypes ranging from cultivated species to wild species. Data presented in the study are convincing and descriptions in text are also appropriate and compact. Comments are below.

Fertility rate (seed formation rate) in tomato plants crossed between MbTMV-MT (MiMe) and Funtelle (MiMe), and between MbTMV-MT (MiMe) and Maxeza (MiMe) should be present, since seed formation rate will be important factor for propagation of seed composed of four haplotypes.

Minor point

Lines 8 from the bottom in Supplemental Note 1: Supplemental Fig. 8 will be Supplemental Fig. 9.

Lines 9 from the bottom in Supplemental Note 1: Supplemental Fig. 7 will be Supplemental Figs. 7 and 8.

Author Rebuttal to Initial comments

Reviewers' Comments:

Reviewer #1:

Remarks to the Author:

The authors describe a system for APH (autopolyploid progressive heterosis) in tomato, by sequentially combining 4 different haplotypes into a tetraploid plant. They utilize the *MiMe* approach that was previously demonstrated to produce unreduced gametes in Arabidopsis and rice. The importance of this system is that it will allow for combinations of a much greater diversity of alleles (including from wild relatives) and more heterotic effects than is possible with conventional breeding.

Their previous *MiMe* design required knockout of the *OSD1* gene, but the authors determined that this was not

possible in tomato, because it is a single copy essential gene in tomato (unlike Arabidopsis and rice). They then show that disruption of the tomato *TAM/Cyclin A1* (also a single copy gene) leads to diploid gametes, similar to *OSD1*. In combination with knockouts of *REC8* and *SPO11*, they get clonal unreduced gametes, with significant pollen lethality but still producing viable seeds that are tetraploid. They then use this approach to make *MiMe* triple knockouts in three different F1 hybrid genotypes, including 2 commercial hybrids. The *MiMe* hybrids makes clonal unreduced gametes, and when cross-pollinated gives rise to tetraploid progeny with unrecombined haplotypes from all 4 parents with high frequency. Plants with the intact genomes of all four haplotypes were obtained and confirmed by whole genome sequencing, that showed normal vegetative growth and produced seedless fruit.

Overall, this study – although submitted as a Brief Communication - is impressively thorough (with 29 Supplementary figures!), and the major results are clearly presented and convincing. Although the system will need improvement before it can be used in actual cultivation, it provides important proof of concept for an approach that opens the door to new breeding schemes that can exploit allelic diversity and heterosis at the polyploid level. Their scheme is potentially applicable to other dicot crops, and as the authors point out, it can be especially useful for crops like potato that can be vegetatively propagated after constructing favourable combinations of parental genomes.

Response: We are glad to read the reviewer's appreciation of our work and their interest in how our findings could be applied in new breeding schemes. We have addressed the specific feedback points below.

Comments:

1. The analysis of mutant plants is focused on the male gametes. A major omission is that there is no information about their effects on the female gametes. It is understandable that detailed characterization of female meiosis and gametes was not undertaken, because it is much more difficult. But indirect inferences can be drawn from their own data. The occurrence of tetraploid progeny implies that some egg cells had to be diploid. This should be stated explicitly, and estimates of the fractions of unreduced clonal female gametes for their different TAM and *MiMe* mutants should be provided, which is important for others who are considering use of this system.

Response: Thank you for this comment. Indeed, we focused on male reproductive development because reproductive phenotyping and cytological analysis is more amenable due to the higher number of meiocytes and gametes. However, we fully agree the lack of quantitative information on unreduced female gametes was a weakness and have endeavored to explore unreduced female gamete frequency by inference by performing crossing experiments using the *Sltam-3* (Micro-Tom background) and *Sltam-4* (Micro-Tom background) mutants we developed in the course of this study. It should be noted that interploidy crosses in plants do not usually lead to full transmission of unreduced gametes due to the triploid block which has been attributed to a failure in endosperm development (either due to maternal or paternal excess) and is well-described in several plant species¹ (Köhler et al., 2009, *Trends in Genetics*). In tomato (*Solanum lycopersicum* previously *Lycopersicon esculentum*) and its wild ancestor (*Solanum pimpinellifolium* previously *Lycopersicon pimpinellifolium*) the triploid block has been formally demonstrated^{2,3} (Jorgensen, 1928, *Journal of Genetics*; Cooper and Brink, 1945, *Genetics*).

Action: We have performed emasculations of wild type (Micro-Tom background), *Sltam-3* (Micro-Tom background) and *Sltam-4* (Micro-Tom background) followed by manual pollination with wild type pollen (Micro-Tom background). To account for possible endosperm failure due to the triploid block, we harvested developing tomato fruits 21 days after pollination, sterilized them and cut them open to assess embryo/seed development and (where possible) rescued developing embryos. Please see Supplementary Figure 10 to see the results of this experiment.

Supplementary Figure 10: Embryo rescue from developing seeds after pollination of *Sltam* mutants with wild type pollen.

- a**, Rescuable and non-rescuable seeds found in Micro-Tom and *Sltam* mutant fruits after manual pollination with wild type pollen from Micro-Tom (MT). The fruits were opened 21 days after pollination. Scale bar= 1mm.
- b**, Quantitative analysis of rescuable and non-rescuable seeds in crossed fruits for wild type (n=48) and *Sltam* (*Sltam-3*, n=153; *Sltam-4*, n=201).
- c**, Quantitative analysis of the outcome of rescued embryos in wild type (n=48) and *Sltam* (*Sltam-3*, n=128; *Sltam-4*, n=153).

Wild type (MT) x wild type (MT) crosses resulted in normal seeds which could be cut open and rescued embryos gave rise to normal diploid plants (MT x MT, 48/48). In contrast, when we cut open the *tam* mutant crosses we already found many non-rescuable seeds (*Sltam-3* x MT, 25/153; *Sltam-4* x MT, 48/201) which we conclude are a result of failed endosperm development due to maternal excess. We rescued the remaining embryos from the *Sltam* crosses and later found those embryos had three different outcomes (viable & diploid; viable & triploid, non-viable & dead). We attribute the sizable “non-viable and dead” class to the triploid block, and conclude that even

though the embryo was rescuable that it was likely already severely malnourished/not properly formed early in plant development. In summary if we attribute all embryo outcomes that are not viable & diploid to the presence of an unreduced female gamete then we could estimate that in *Sltam-3* (70/153) 46% and in *Sltam-4* (111/201) 55% of female gametes are unreduced. More cautiously, some of the failed embryo developments could be due to reasons other than the triploid block so a conservative estimate may suggest a female unreduced gamete frequency of between 30-50%. This number would be consistent with estimates for male gametes where all five *tam* lines produced pollen where 29-48% of pollen was unreduced (Figure 1c).

Further corroboration of these numbers is found from the ploidy of *Sltam* mutant selfing offspring. We found that from *Sltam-3* selfing seed we retrieved 5 tetraploid plants out of 73 offspring, and for *Sltam-4* we retrieved 7 tetraploid plants out of 66 offspring (Extended Data Fig.1f). This is a cumulative tetraploidy rate of 9.3% which could be explained by 30% male unreduced gametes and 30% female unreduced gametes ($0.3 \times 0.3 \times 100 = 9\%$).

2. To follow up on the previous comment, Extended Data Fig. 1 shows tetraploid frequencies of 10-15% for two TAM mutants, whereas main Fig. 1h shows 93% tetraploid frequency for a *MiMe* mutant. Why would the tetraploid frequency increase? Is it because mutations in *SPO11* and *REC8* are lethal unless rescued by loss of TAM function? If so, this would account for the very low seed set, because low penetrance of the TAM mutation during female meiosis would lead to mostly dead seeds, the only survivors would be tetraploid.

Response: Thank you for this point. As previously indicated we found that in *Sltam-3* mutants, we got 5 tetraploid plants (6.85%) from 73 plants grown from selfing seeds, and for *Sltam-4* mutants, we got 7 tetraploid plants (10.61%) from 66 plants from selfing seeds. In contrast the offspring of *MiMe* plants are tetraploid at a much higher penetrance (Figure 1h and Extended Data Fig.2e). The reason for the higher penetrance of tetraploidy in the *MiMe* offspring is indeed likely due to the lethality of gametes that have progressed through meiosis I without *SPO11-1* and *REC8* and enter meiosis II (due to incomplete penetrance of skipping entry to meiosis II in *Sltam*). Our results in tomato (Extended Data Fig.2b) and previous results in Arabidopsis⁴ (d'Erfurth et al., 2009, *PLoS Biology*) demonstrated that *spo11-1 rec8* double mutants are highly sterile and fertility is only possible when combined with a mutation that skips entry to meiosis II.

3. Since the authors have extensively used pollen size as a convenient proxy for ploidy, they should provide a reference where correlation between size and ploidy was established (preferably for tomato pollen). Can the possibility that some larger pollen grains are aneuploid be excluded?

Response: In the original submission we included a reference where Arabidopsis pollen from plants with different ploidies (diploid, tetraploid, octoploid) were used as a control⁵ (De Storme, N., et al., 2013, *Plant Reproduction*); this reference also included a demonstration of using this approach to study pollen size from a diploid tomato plant. We have also now added a reference from a recent paper where frequencies of reduced and unreduced tomato pollen are identified using the same approach (Multisizer 4e)⁶ (Schindfessel et al., 2023, *Frontiers in Plant Science*), where

meiotic cytology has also been carried out.

Regarding the second point, there is a possibility that a small percentage of larger pollen grains are aneuploid. however, we find that highly aneuploid gametes that are produced in the tomato *Sspo11-1* and *Srec8* mutant are not viable and have a smaller size (we find that tomato pollen with a particle diameter less than 20 μm is not viable).

Wild type Micro-Tom Pollen

spo11-1 mutant pollen (Micro-Tom background)

rec8 mutant pollen (Micro-Tom background)

4. The *tam-6* allele appears in multiple figures, but its origin is buried in Supp. Table 2. It should be mentioned up front, in the Supplementary Note.

Response: Thank you for bringing this to our attention. To make it clearer we have now explained the origin of *Sltam-6* allele in Supplementary Note 1.

5. Fig. 1c shows that *MiMe* in the commercial hybrids results in high pollen lethality, which is not observed with *MiMe* in MT. Can the authors comment on this difference? E.g. Is the *tam* mutation less penetrant in the hybrids?

Response: Thanks for pointing this out. We agree that the pollen viability varies in the *MiMe* triple mutants in different genetic backgrounds (inbred Micro-Tom and three different hybrid lines). One explanation is that the genetic background modifies the penetrance of the *MiMe* system, which has been previously observed in rice *MiMe* mutants. For example, rice *MiMe* mutants that have been generated in Chunyou84 (CY84), Hwayoung and Nipponbare backgrounds exhibit different levels of fertility^{7,8} (Mieulet et al., 2016, *Cell Research*; Wang et al., 2019, *Nature Biotechnology*). All mutant alleles present in the three hybrid lines we studied in greatest detail (Maxeza F1, Funtelle F1 and Moneyberg-TMV x Micro-Tom F1) are predicted to result in loss of function alleles however we cannot fully exclude the possibility that some *SITAM* mutants may have some residual function.

6. Page 2 lines 12-13: “MbTMV-MT *MiMe* mutants produced smaller fruits that contained less seeds (Fig.1 e,f)”. Should say “much fewer seeds” based on Fig. 1f.

Response: Thank you for raising this point. We agree and have amended the text to read “much fewer seeds”.

7. It would be helpful if the authors could discuss why deletions and truncations sometimes arise in their *MiMe* plants, and to what extent these are likely to pose a problem when scaling up for practical applications.

Response: From the whole genome sequencing results, we identified some tetraploid plants that lost chromosome regions (three copies were present rather than four). To explain this, on page 2, line 25 we added the sentence:

“These chromosome truncations may arise due to *SPO11-1* independent DNA DSBs (e.g., arising from DNA replication stress or environmental DNA damage), which could not be repaired by homologous recombination due to the absence of homologous chromosome pairing and disturbed sister chromatid cohesion”.

We expect that this phenomenon would be of less importance when dealing with clonally propagated plant materials

as is the case in potato (as one could screen for these events by phenotyping and/or sequencing as we have done here), however, if seeds should be used for propagation it would be advantageous to develop a *MiMe* system with higher penetrance.

8. For the analysis of read coverage/SNP changes, they have used a 1 Mb sliding window, which may not give enough resolution to find smaller deletions or truncations. The authors should mention this issue in the Supplementary Note.

Response: Thank you for this point. We previously tested smaller window sizes and decided to use 1 Mbp sliding window as this gave the best signal to noise ratio. There are many structural variations in the 0-1 Mbp size category hence we found reliable detection of smaller deletions/truncations is not robust.

9. Suppl. Fig. 20 could use a clearer and more explanatory legend summarizing the conclusions from the data presented here. Also, the colours are difficult to distinguish, especially in 20b. The other figures similar to this one would also benefit from more explanatory legends, as would Suppl. Figures 22-23 (e.g., what do they conclude from these figures).

Response: Thank you for this suggestion. We modified the Suppl. Fig. 20, 22-23 (now Suppl. Fig. 24, 26-27) legends so that they contain more explanation and context. Regarding the Figure 20 and colours: the purpose here is to see the line trends to infer presence/absence of recombination/aneuploidy, and not to facilitate detailed examination of each individual sample. Most lines are overlapping, which is expected from the non-recombinant tetraploid samples.

10. Suppl. Fig. 29: Though it is mentioned in the Methods section, the figure legend should additionally include the significance of the Tm-2 introgression, for the benefit of researchers not familiar with tomato genetics.

Response: Thank you for this comment - we have added further explanation regarding TMV resistance provided by the Tm-2² introgression to this figure legend (now Supplementary Figure 33).

11. Page 3 line 24 – What is meant by “harness heterosis – on farm – at the polyploid level”? Do they mean “harness heterosis at the polyploid level” for farming/agriculture?

Response: Thank for this point. We have modified the text so that it now reads “harness heterosis at the polyploid level on the farm”.

Reviewer #2:

Remarks to the Author:

In this manuscript by Wang and colleagues, the authors set out to apply the “*Mitosis instead of Meiosis*” (*MiMe*) system that has been previously published in Arabidopsis and the monocot crop rice to the dicot crop tomato. To establish the *MiMe* system in tomato, the authors first identify homologs of *AtOSD1*, *AtREC8*, and *AtSPO11-1*, the genes which have been mutated for *MiMe* in Arabidopsis. The authors find that mutants lacking *SOSD1* activity are lethal, and explore another mutation that has been previously shown to skip the meiotic division in Arabidopsis (*tam1*). The authors then demonstrate in an elegant way that the *MiMe* system with *SISPO11-1*, *SIREC8*, and *SITAM* allows the production of unreduced clonal gametes in tomato and the “fixation” of hybrid genotypes. Furthermore, the authors cross distinct *MiMe* hybrid genotypes and produce viable tetraploid offspring that combines haplotypes from 4 distinct parents.

The application of the *MiMe* system in a dicot crop and the combination of agronomically-important alleles from 4 distinct haplotypes by hybridization of clonal gametes from two hybrids is novel. However, whether heterosis effects can be conserved through the *MiMe* approach remains unclear. The authors use state-of-the-art technologies to produce high-quality data, which leads them to mainly sound conclusions. In parts, the manuscript lacks clarity, mainly because of its density and the organization of the Suppl. Data. I have several comments about the presentation and interpretation of the data, which I outline in detail below.

Response: We thank the reviewer for their comments and are glad that they appreciate the novelty of applying *MiMe* in polyploid breeding. Please see below for our response to the specific points.

Major comments

page 1, line 43-46: The authors report that they established the *MiMe* system in tomato by mutating the three genes *SISPO11-1*, *SIREC8*, and *SITAM*. They refer in a single sentence to 1 Supplementary Note, 3 panels of Fig. 1, 2 Extended Data Figures, and 15 Suppl. Figures. While their work on *SIOSD1* and *SITAM* is described in more detail in the Suppl. Note, their data and findings on *SISPO1* and *SIREC8* remains largely un-commented. The authors should describe the data in Suppl. Fig. 10-15 and their conclusions either in the Main Text or Suppl. Note. I also suggest to include the two Extended Data Figures to the list of Supplementary Figures to make the manuscript easier accessible to the reader.

Response: Thank you for this suggestion. We have now included a detailed description regarding the *Slspo11-1* and *Slrec8* single mutants in Supplementary note 1. The text we have included is:

“To further confirm the conserved meiotic function of *SPORULATION 11-1* (*SPO11-1*) and *REC8* in tomato, we obtained two independent alleles of *Slspo11-1* and *Slrec8* mutants in Micro-Tom background (Supplementary Table 2, Supplementary Fig.11 and Supplementary Fig.12). To explore the reasons for complete male sterility in both single mutants (Extended Data Fig.2b), we carried out observations of chromosome behavior in male meiocytes. In *Slspo11-1-1* mutants, an abnormal meiotic process was observed where twenty-four univalent were present instead of twelve bivalents at diakinesis, indicating no meiotic recombination (Supplementary Fig.14). The random segregation of homologous chromatids during meiosis I and subsequent segregation of sister chromatids during meiosis II resulted in aborted polyads (Supplementary Fig.14). In *Slrec8-1* mutants, there is no typical pachytene stage due to defective homologous chromosome pairing (Supplementary Fig.15). At diakinesis, abnormal tangled chromosomes with partial chromosome bridges occurred. Subsequently we observed more than 12 chromosomes randomly segregating towards the two opposite poles at anaphase I, indicating premature separation of sister chromatids at meiosis I (Supplementary Fig.15). In *Slspo11-1 Slrec8* double mutants, the first meiotic division mimics the mitotic cell division leading to the balanced segregation of sister chromatids and then the second division is unbalanced resulting in aneuploid gametes (Supplementary Fig.16).”

We have not decided to not change the Extended Data Figures 1 and 2 into supplementary figures. We understand this arrangement is more complex for the reviewer, however this data is quite central to establishing the *MiMe* system that is applied in Figures 1 and 2. If the figures are included as Extended Data Figures, we understand that they will be included in the .pdf version of the paper and therefore more readily accessed for online readers (compared with the Supplementary Figures which would be accessed separately).

page 2, line 3: The authors write that they “identified six MbTMV-MT, three Funtelle and three Maxeza lines that had biallelic 4 mutations on *SISPO11-1*, *SIREC8*, and *SITAM* (Supplementary Table 4).” It is unclear from the main text and methods how the mutations were induced (multiplex, number of gRNAs, etc.). In addition, Suppl. Table 4 shows only data for 4 MbTMV-MT lines, one Funtelle, and one Maxeza line. The MbTMV-MT lines A, C, and D show identical alleles for all three genes suggesting that they descend from a single transformation event and are not independent events. The authors should clarify.

Response: Thank you for bringing this to our attention. We developed two constructs that target *SPO11-1*, *REC8* and *TAM*. The plasmids are listed in Supplementary Table 1 and were called pYZ1 and pYZ182. pYZ1 has two guide RNAs per gene and was developed first and used in the Micro-Tom work as listed in Supplementary Table 1. For the hybrid lines we developed a new plasmid (pYZ182) containing the most efficient guide RNAs for each gene and it contains in total three gRNAs (one guide RNA per gene) as listed in Supplementary Table 1.

We have added a clarification below the table stating: “pYZ1 has two guide RNAs per gene (2 gRNAs Vs *SPO11-1*, 2gRNAs Vs *REC8*, 2 gRNAs Vs *TAM*) and was used in the work in inbred Micro-Tom. pYZ182 has one guide RNA

per gene (1 gRNA Vs *SPO11-1*, 1 gRNA Vs *REC8*, 1 gRNA Vs *TAM*) and was used in the work on MbTMV- MT F1, Maxeza F1 and Funtelle F1.”

We have also further examined the mutations in the lines we identified in MbTMV-MT, Funtelle and Maxeza. The result of this is that we now refer in the paper to two independent lines in MbTMV-MT, three in Funtelle and three in Maxeza. We have modified the statement in the paper to read:

“identified two independent MbTMV-MT, three independent Funtelle and three independent Maxeza lines that had biallelic mutations on *SISPO11-1*, *SIREC8*, and *SITAM* (Supplementary Table 4)”

We apologize that this was not clear in the wording of the previous version.

page 2, line 13-14: The authors report that the MbTMV-MT *MiMe* plants produced smaller fruits with larger seeds. Quantitative data for the seed size increase should be presented to support the claim.

Response: Thank you for this comment. We have now performed a quantitative analysis of the seeds collected from MbTMV-MT hybrid control and MbTMV-MT^{MiMe} plants. This validates that the seeds collected from *MiMe* plants are bigger. Please see the Supplementary Figure 18 which is also pasted here for your convenience.

Supplementary Figure 18: Quantitative seed size analysis of seeds collected from MbTMV-MT F1 hybrid and MbTMV-MT^{MiMe_A}.

a, dry mature seeds from MbTMV-MT F1, scale bar= 1mm. **b**, dry mature seeds from MbTMV-MT^{MiMe_A}, scale bar= 1mm. **c**, Seed size quantitative analysis in MbTMV-MT F1 hybrid and MbTMV-MT^{MiMe_A}. Seed images were taken using LAS X software and processed using the "threshold" feature of ImageJ (<https://imagej.net/software/fiji/downloads>). Seed size area was measured using the "Analyze Particles" feature,

with a lower limit “1-Infinity mm²” to exclude any non-seed material.

page 2, line 23-24: The authors write that they “validated the partial loss of one of the MbTMV copies of chromosome 9 (Fig. 1i).” However, the panel of Fig.1i rather shows the observation of the loss based on allele frequency data. The authors should either change their wording or validate the partial loss using an alternative approach.

Response: Thank you for this point – we accept it was not fully clear and have amended the wording so the figures are referred to properly. In Fig 1i we indeed just use the allele frequency information. In Supplementary Figure 20 we show 2 “normal samples” without truncation (MbTMV-MT F1#1 and MbTMV-MT^{MiMe-A} #1) and one sample with truncation (MbTMV-MT^{MiMe-A} #3). In that figure we show both the allele frequency information and the normalized read coverage. As we state in the legend of Supplementary figure 20 we agree with the reviewer that both conditions (diverged allele frequency and reduced coverage) should be satisfied to distinguish true chromosome truncation from genetic deletions in the Micro-Tom genome relative to the MbTMV reference.

page 2, line 38: The authors write that they generated a platinum-grade assembly for Micro-Tom. Assembly statistics should be included in the manuscript to allow the reader to assess the quality of the assembly.

Response: Thank you for this comment and the suggestion to share the Micro-Tom genome we have assembled as part of this work. We have now expanded on the description of the Micro-Tom genome assembly that we had in the methods by now including a Supplementary Note (Supplementary Note 2) on this topic. Further information on the assembly process including raw read K-mer analysis, HiC plot and genome alignment to Mb-TMV genome are presented in a new Supplementary Figure 22 (see below). We have also added raw hifiasm assembly statistics (Supplementary Table 5), scaffolded assembly statistics (Supplementary Table 6) and raw-read coverage analysis of the final genome (Supplementary Figure 23). The raw data for the Micro-Tom genome assembly is made available at the European Nucleotide Archive (PRJEB62441) and the final assembly is made permanently available at datadryad.org (<https://doi.org/10.5061/dryad.h9w0vt4qd>).

Supplementary Figure 22. Chromosome scale *de novo* genome assembly of Micro-Tom.

a, K-mer Analysis Toolkit (KAT) results of the K-mer distribution in Micro-Tom HiFi data. *S. lycopersicum* cv. Micro-Tom HiFi K-mer frequency distribution where $K = 21$. **b**, Hi-C contact plot of Micro-Tom hifiasm assembly scaffolded with Omni-C data. Hi-C contact plots are created by Salsa2 and Juicebox. Vertical and horizontal lines represent scaffold borders. **c**, *S. lycopersicum* cv. Micro-Tom genomes aligned against *S. lycopersicum* cv. Moneyberg-TMV (van Rengs et al., 2022). Dotplots are generated using D-genies (Cabanettes and Klop, 2018). Horizontal grey dashed lines represent MbTMV chromosome borders. Vertical grey dashed lines represent Micro-Tom chromosome borders. Unplaced contigs aligned to Moneyberg-TMV “ch00” are marked by a red line.

**Supplementary Figure 23. Characterization and validation of the 12 Micro-Tom chromosomes.**

Characterization and validation by read coverage analysis in 100kb windows plotted over the genomic position on the genome (Red solid lines represents HiFi read coverage). Dashed horizontal line represents mean coverage. Black circles represent coverage outliers (<2.5% and >97.5% percentiles). X-axis is chromosome position in Mbp.

Genotype	Micro-Tom	Funtelle		Maxeza	
Haplotypes	N/A	Haplotype 1	Haplotype 2	Haplotype 1	Haplotype 2
Contigs	2739	1155	651	1488	689
Total length (Mb)	925.2	875.2	839.5	869.4	846.2
Max length (Mb)	52.6	76.12	69.815	73.183	74.75
N50 length (Mb)	21.1	56.5	41.3	55.7	39.4
N90 length (Mb)	0.077	1.3	1.8	2.1	4.6
L50	14	7	8	7	8
L90	229	29	34	25	27

'N/A' means 'Not applicable'

Supplementary Table 6. Micro-Tom and MbTMV scaffolded chromosome statistics and quality metrics.		
Accession	Micro-Tom	Moneyberg-TMV
Species	S. lycopersicum	S. lycopersicum
Reference	This study	van Rengs et al., 2022
Number of sequences	12	12
Number of sequences (>50kb)	12	12
Cumulative size (Mbp)	812.44	824.45
N50 (Mbp)	67.8	68.5
N90 (Mbp)	56.8	54.7
L50	6	6
L90	11	11
Longest sequence (Mbp)	96.3	96.5
Number of N's	4000	0
Number of internal N-regions	40	0
Raw LAI	8.42	9.67
LAI	14.05	15.3
Complete BUSCO (C)	5851	5852
Complete Single copy (S)	5743	5747
Complete Duplicated (D)	108	105
Fragmented (F)	12	12
Missing (M)	87	86
Total searched (solanales) busco v5.2.1	5950	5950
QV	72.40	54.12
Completeness	99.22	99.10

Fig. 2f: The Figure shows major structural variation on chromosome 6 between the two Funtelle haplotypes. However, it is unclear how the data supports the presence of Mi resistance variant. Please explain.

Response: To support the presence of root knot nematode resistance gene *Mi-1* (Solyc06g008790), we visualized the presence of a *Mi-1*-linked 56-bp deletion^{9,10} (Garcia et al., 2007; Devran et al., 2016) in a new supplementary figure (Supplementary Figure 32, see below). From that published marker we can show the Funtelle variety has the resistance gene by checking the sequencing data (Supplementary Figure 32, see below). In addition we can find in the Syngenta seed brochure that Funtelle has root knot nematode resistance by Mi: (<https://www.syngenta.nl/product/seed-vegetable/tomaat/funtelle>).

Supplementary Figure 32: IGV visualization of *Mi-1* and *Tm-2²* introgression in control and 4-Hap plants.

a, The Integrative Genomics Viewer (IGV) visualization of *Mi-1*-linked 56-bp deletion (Garcia et al., 2007; Devran et al., 2016). Funtelle contains heterozygous introgression of *Mi-1* indicated by the haplotype spanning both the deletion and the SNPs upstream the deletion. **b**, IGV visualization of *Tm-2²*, plotted with the primers discriminating resistant and susceptible alleles (Lanfermeijer et al., 2005). Micro-Tom and Funtelle has no and heterozygous *Tm-2²* introgression, respectively, while both MbTMV and Maxeza have homozygous introgression. SNPs within the red box in **a** and **b** were used to genotype the 4-Hap samples. **c**, Percentage of matched bases (relative to gene size) between the parental assembly and the *Tm-2²* and *Tm-2* (susceptible) sequences.

page 3, line 6-7 and Fig. 2g: It is unclear how the authors conclude from Fig. 2g that “MbTMV-MT-Maxeza^{MiMe} plants have more Tomato Mosaic Virus (TMV) resistance alleles than MbTMV-MT-Funtelle^{MiMe}” or whether Fig. 2g supports another conclusion.

Response: Thank you for this comment – it has triggered us to perform extra analysis which we think strengthens this part of the paper considerably. Previously the graph we presented as 2e (gene dosage of agronomically relevant genes in 4-Hap plants) was entirely based on the sequences of the parents (this is retained in the new figure 2e as the grey histogram). Now we used the whole genome sequencing data of the actual 4-Hap plants and genotyped each 4-Hap plant for each gene and plotted allele frequency for each plant over the histogram (see new plot below). We find close agreement between the predicted genotype (grey histogram) with the actual genotypes (box and whiskers).

Fig.2e, The expected gene dosage in 4-Hap plants (histogram) and WGS-based genotyping of 4-Hap plants (box and whisker plot) regarding tomato mosaic virus resistance (*Tm-2²*, Solyc09g018220), *Meloidogyne incognita* (*Mi*, Solyc06g008720), self-pruning (*SP*, Solyc06g074350), dwarf (*D*, Solyc02g089160) and *I* gene (Solyc11g011180).

In our opinion the results of 2f and g are better discussed subsequent to the description of 2e so we have changed the text to read as follows:

“We predicted agronomically relevant gene dosage in 4-Hap plants based on parental genome sequences and subsequently counted allele frequency in the 4-Hap plants themselves (Fig.2e). This revealed that genotypes were inherited as expected if gametes were clonal (Fig.2e). Marker, synteny and pangenome analysis confirmed the introgression of *Meloidogyne incognita* (*Mi*) resistance in Funtelle haplotype-1 (Fig.2f,g and Supplementary Fig.32). The genotyping analysis showed that MbTMV-MT-Maxeza^{MiMe} plants have more copies of the *Tm2²* haplotype than MbTMV-MT-Funtelle^{MiMe} (Fig.2e and Supplementary Fig.32), as predicted from the parental genome sequences.”

Minor comments:

page 1, line 46: The authors explain that they implemented the *MiMe* system in three agronomically-relevant hybrid genotypes. One of those is a Moneyberg-TMV x Micro-Tom hybrid. Can the authors explain the agronomic relevance of this hybrid given that Micro-Tom is mainly a model for basic research?

Response: Thanks for this point. We have modified this statement to read:

"We implemented the *MiMe* system in three hybrid tomato genotypes including the Moneyberg-TMV x Micro-Tom (MbTMV-MT) model hybrid, the date-tomato commercial hybrid "Funtelle" and the truss tomato commercial hybrid "Maxeza".

page 2, line 12. The authors find that the MbTMV-MT *MiMe* plants develop smaller fruits than the MbTMV-MT parents. Can the authors speculate why fruits are smaller and how this negative effect could be overcome?

Response: Indeed, we have found that MbTMV-MT^{*MiMe*} plants develop smaller fruits than wild-type MbTMV-MT control. We speculate that this reduced fruit size could be related to the lower seed number in the fruits of MbTMV-MT^{*MiMe*} plants and as a result of less seed setting the reduced production of hormones in the developing fruits. If the seed production is the key factor in fruit size in this case it could be useful develop a *MiMe* system in tomato with higher penetrance, which we think would require an alternative mutant to more efficiently skip the second meiotic cell division (*tam* is not full penetrance)

page 3, line 9. The authors write that the 4-H plants developed "well-organized branches with seedless fruits". The authors should specify whether they refer to reproductive branches (inflorescences) or vegetative branches (shoots).

Response: Thanks for this point. Both are normal but we have clarified this sentence to read: "and the production of well-organized inflorescences that harbor seedless fruits"

In addition, the authors could include images of the parental genotypes in Fig. 2 and Suppl. Fig. 25 as comparison to allow the reader to determine whether development of shoots, inflorescences, and fruits are abnormal in the 4- H plants.

Response: We have added plant images of MbTMV-MT^{*MiMe*}, Maxeza^{*MiMe*} and Funtelle^{*MiMe*}, MbTMV-MT F1, Maxeza F1 and Funtelle F1, and fruit images of MbTMV-MT F1, Maxeza F1 and Funtelle F1 in Supplementary Figure 34.

Supplementary Figure 34: Plant morphology of hybrid *MiMe* plants and F1 hybrid plants.

a, Young plant morphology of hybrid *MiMe* plants and control F1 hybrid plants (From left to right: MbTMV-MT^{MiMe}, Maxeza^{MiMe}, Funtelle^{MiMe}, MbTMV-MT F1, Maxeza F1 and Funtelle F1).

b, Selfing mature fruits of three F1 hybrid plants (MbTMV-MT F1, Maxeza F1 and Funtelle F1). Scale bars= 2cm.

page 13, line 1: The authors should make all assemblies that were newly generated in this study available to the community.

Response: The genome assemblies of hybrid varieties Funtelle and Maxeza (<https://doi.org/10.5061/dryad.931zcrjs4>), and the genome assembly of inbred Micro-Tom (<https://doi.org/10.5061/dryad.h9w0vt4qd>) are made available at datadryad.org via the respective links.

Figure 1a: The schematics of tomato plants suggest that tomato plants develop fruits and flowers on their compound leaves, which could lead to botanical misconceptions.

Response: We agree and have modified the schematic image as such for Fig.1a.

Fig.1a, Schematic workflow of the generation of *MiMe* (*spo11-1 rec8 tam*) triple mutants in four tomato genotypes.

Reviewer #3:

Remarks to the Author:

This study uses the *MiMe* system of bypassing meiosis to produce tetraploids with four distinct genomes that would avoid the problems of meiotic disturbances in autotetraploids and that would be uniform. Producing a double cross tetraploid by sexual means will introduce variability in a quadruplex hybrid due to double reduction in the hybrid tetraploid parents and aneuploid progeny from segregations that are not 2:2 because of different types of associations of the four homologues in meiosis. The authors have achieved their goal. Some thoughts on the presentation follow.

Response: Thank you very much for your comments and appreciation of our work.

The manuscript is very densely written and could probably be expanded with greater explanations that would be helpful to the more general reader. It is classified as a Brief Communication but perhaps the editors could allow an expansion.

The authors use the term 4-H to designate a tetraploid with four distinct genomes. The classical term for this configuration is a quadruplex tetraploid. It is fine to use the 4-H designation, but the authors should be aware of the priority term. (The authors should also be aware that "4-H" is a youth organization in the United States.)

Response: Thank you for bringing this to our attention. In light of this information, we decided to change the term of the "4-H" plants to "4-Haplotype" plants which we shorten to "4-Hap" plants. We rephrased this part of the paper as

pasted below:

“According to classical nomenclature such plants could be referred to as “non-recombinant quadruplex hybrids” but for simplicity we refer to them as “4-Haplotype” (4-Hap) plants^{2,8}”. Here we added relevant citations to a review (2) and a research article (8) on this topic:

2. Washburn, J. D. & Birchler, J. A. Polyploids as a ‘model system’ for the study of heterosis. *Plant Reprod.* 27, 1–5 (2014).

8. Riddle, N. C. & Birchler, J. A. Comparative analysis of inbred and hybrid maize at the diploid and tetraploid levels. *Theor. Appl. Genet.* 116, 563–576 (2008).

In Figure 1j, it is difficult to distinguish between the F1 and F2 lines. The green and blue lines appear very similar. The authors might consider a more distinctive color difference.

Response: Thank you for this point. We realized that the blue line was a dashed line and have now modified the Figure 1j, so it has only solid lines that we hope makes it possible to more easily distinguish the F1 and F2 populations.

In the Introduction, the authors note that the system could help capitalize on progressive heterosis and this reviewer agrees that this procedure would be very beneficial for making it more uniform. But it was not clear whether there was progressive heterosis in the 4-H material produced. In Figure 2h there is no comparison to 2-H tetraploids and the data in Figure 2i do not make it clear. Perhaps there is midparent heterosis? At any rate, by including MicroTom, the biomass characters might be too skewed for progressive heterosis to be obvious. When the four grandparents are more comparable, then it is more obvious.

Response: We fully appreciate this important point. The main novelty of our paper is that by introducing *MiMe* in hybrid crop genotypes we can generate clonal gametes and then use this to precisely engineer polyploid genomes (by polyploid genome design). We think we have satisfied the reviewers that this is the case.

One of the reasons polyploid genome design is interesting (as you have pointed out) is it could allow controlled exploitation of autotetraploid progressive heterosis (APH) in hybrid crops. As pointed out we did not perform a side-by-side analysis of F1 hybrid (diploid), 2-Hap (tetraploid) and 4-Hap (tetraploid) plants in this work. This would be required for us to claim autotetraploid progressive heterosis, which we do not. We have clarified this now in the discussion by including the lines:

“In this report we demonstrate that clonal gamete production in hybrid crop genotypes allows precise polyploid genome engineering, yet the exploitation of autopolyploid progressive heterosis will involve further steps to be taken. This will require the development of four-way heterotic groups, which could be driven by using genomic selection to identify higher-order combining abilities between grandparental lines”.

Regarding the involvement of Micro-Tom in the crosses: we agree that Micro-Tom has some downsides with respect to finding evidence for APH in Tomato. Micro-Tom is a dwarf tomato variety that was developed during the 1980s. As a result of including it in the plant materials leading to the 4-Hap plants we thought it was better to not focus on heterosis testing, which would be more meaningful to perform on modern greenhouse tomato

cultivars with specific combining abilities. Despite this downside of including Micro-Tom, we suspect that the inclusion of Micro-Tom may be important for parthenocarpic fruit development in the 4-Hap plants. In our Micro-Tom *spo11-1* and *rec8* mutants, we observed the formation of small seedless fruits at high penetrance that we believe are parthenocarpic (Extended Data Figure 2a). Future planned work in our laboratory aims to understand whether or not the Micro-Tom genetic background is required for the high penetrance of parthenocarpic fruits in 4-Hap plants. In addition, as alluded to in our discussion, we think that potato may be a more suitable model system to further explore APH and polyploid genome design via *MiMe* as heterosis is more notable in potato than tomato. Overall, we think the heterosis testing is important but it is beyond the scope of the current study due the plant materials we currently have available to us.

On page 3 lines 26-27: cultivated strawberry is an allopolyploid (Nature Genetics, 2019, 51: 541-547). The seedless nature of banana has as much to do with parthenocarpy as to its triploid nature. Some bananas have three genomes of *Musa acuminata* while others are triploid hybrids of *M. acuminata* and *M. balbisiana* (See references in D'Hont et al 2012, Nature 488: 213-217). The sentence might be taken to suggest that they are all autopolyploids.

Response: Thank you for this point. We have modified this sentence so that it broadly refers to polyploids rather than autopolyploids specifically. The sentence now reads:

“Taking a wider perspective, polyploid genome design could be employed for the clonal transfer of genomes from diploid wild materials into current polyploid crops (e.g., strawberry) and the generation of highly heterozygous seedless triploid varieties (e.g., banana).”

Reviewer #4:

Remarks to the Author:

MiMe (*Mitosis instead of Meiosis*) established in rice and *Arabidopsis* has provided the chance of transmission of hybrid genome into unreduced gametes without recombination. In addition to *MiMe*, parthenogenetic development of unreduced egg cell triggered by ectopic expression of an AP2-type transcription, such as *OsBBML1*, is required for formation of clonal rice seeds (progenies). In the present manuscript (NG-BC62972R), the authors introduced *MiMe* into tomato to establish the artificial building-up system of autopolyploid progressive heterosis (APH). Although *MiMe* introduction into tomato is not new/unique concept, the authors successfully combined haplotypes of interests without recombination in tomato. These *MiMe* based APH in tomato may provide a horizon for utilization of haplotypes ranging from cultivated species to wild species. Data presented in the study are convincing and descriptions in text are also appropriate and compact. Comments are below.

Response: Thank you very much for your positive comments on our work.

Fertility rate (seed formation rate) in tomato plants crossed between MbTMV-MT (*MiMe*) and Funtelle (*MiMe*), and between MbTMV-MT (*MiMe*) and Maxeza (*MiMe*) should be present, since seed formation rate will be important factor for propagation of seed composed of four haplotypes.

Response: Thank you for your good suggestion. We have presented the related data for fruit sizes and seed number for crosses between MbTMV-MT^{*MiMe*} and Funtelle^{*MiMe*} and crosses between MbTMV-MT^{*MiMe*} and Maxeza^{*MiMe*} in Supplementary Table 8.

Supplementary Table 8. Seed formation rate after hybridization of hybrid MiMe and control plants.				
	Female	Male	Fruite weight (g)	Seed number
Control	Maxeza F1	MbTMV-MTV F1	79.78	68
			61.19	59
			68.38	49
			63.39	58
			54.11	57
	Funtelle F1	MbTMV-MTV F1	9.84	45
			7.79	38
			5.83	23
			6.81	31
			6.74	19
4-hap	Maxeza ^{MiMe}	MbTMV-MTV ^{MiMe}	24.63	5
			17.52	3
			14.17	2
			29.96	6
			24.43	4
	Funtelle ^{MiMe}	MbTMV-MTV ^{MiMe}	6.02	3
			5.89	2
			4.51	1
			4.33	1
			4.97	1

Minor point

Lines 8 from the bottom in Supplemental Note 1: Supplemental Fig. 8 will be Supplemental Fig. 9.

Lines 9 from the bottom in Supplemental Note 1: Supplemental Fig. 7 will be Supplemental Figs. 7 and 8.

Response: Thank you for these suggestions. We modified the figures we referred to in Supplemental Note 1 accordingly.

References:

1. Köhler, C., Mittelsten Scheid, O. & Erilova, A. The impact of the triploid block on the origin and evolution of polyploid plants. *Trends Genet.* **26**, 142–148 (2010).

2. Jorgensen, G. A. The experimental formation of heteroploid plants in the genus solanum. *J. Genet.* **19**, 133–210 (1928).
3. Cooper, D. C. & Brink, R. A. Seed Collapse following Matings between Diploid and Tetraploid Races of *Lycopersicon Pimpinellifolium*. *Genetics* **30**, 376 (1945).
4. d'Erfurth, I. *et al.* Turning Meiosis into Mitosis. *PLoS Biol.* **7**, e1000124 (2009).
5. De Storme, N., Zamariola, L., Mau, M., Sharbel, T. F. & Geelen, D. Volume-based pollen size analysis: An advanced method to assess somatic and gametophytic ploidy in flowering plants. *Plant Reprod.* **26**, 65–81 (2013).
6. Schindfessel, C., De Storme, N., Trinh, H. K. & Geelen, D. Asynapsis and meiotic restitution in tomato male meiosis induced by heat stress. *Front. Plant Sci.* **14**, 1210092 (2023).
7. Mieulet, D. *et al.* Turning rice meiosis into mitosis. *Cell Res.* **26**, 1242–1254 (2016).
8. Wang, C. *et al.* Clonal seeds from hybrid rice by simultaneous genome engineering of meiosis and fertilization genes. *Nat. Biotechnol.* **37**, 283–286 (2019).
9. Garcia, B. E. *et al.* A co-dominant SCAR marker, Mi23, for detection of the Mi-1.2 gene for resistance to root-knot nematode in tomato germplasm. (2007).
10. Devran, Z., Göknur, A. & Mesci, L. Development of molecular markers for the Mi-1 gene in tomato using the KASP genotyping assay. *Hortic. Environ. Biotechnol.* **57**, 156–160 (2016).

Decision Letter, first revision:

11th Jan 2024

Dear Dr Underwood,

Your Brief Communication, "Harnessing clonal gametes in hybrid crops to engineer polyploid genomes" has now been seen by 4 referees. You will see from their comments below that while they find your work of interest, some important points are raised by Reviewers 1 & 2. We are interested in the possibility of publishing your study in Nature Genetics, but would like to consider your response to these concerns in the form of a revised manuscript before we make a final decision on publication.

We therefore invite you to revise your manuscript taking into account all reviewer and editor comments. Please highlight all changes in the manuscript text file. At this stage we will need you to upload a copy of the manuscript in MS Word .docx or similar editable format.

*2) If you have not done so already please begin to revise your manuscript so that it conforms to our Brief Communication format instructions, available here.

*3) Include a revised version of any required Reporting Summary:

Please be aware of our guidelines on digital image standards.

[redacted]

We hope to receive your revised manuscript within four to eight weeks. If you cannot send it within this time, please let us know.

Sincerely,
Wei

Wei Li, PhD
Senior Editor
Nature Genetics
New York, NY 10004, USA
www.nature.com/ng

Reviewers' Comments:

Reviewer #1:

Remarks to the Author:

The authors have addressed all my comments satisfactorily. There is just one point which needs elaboration in the manuscript: The estimated frequencies of unreduced female gametes in TAM mutants have been provided in the Rebuttal, but I could not find numbers for these estimates in the manuscript. They should be explicitly stated in the Supplementary Note 1, in the paragraph following the sentence "We also measured unreduced female gamete frequency by inference by performing crossing experiments using the Sltam-3 and Sltam-4 mutants." Other than this, the manuscript can be accepted without further revisions. It represents a significant contribution of high interest to plant genetics and breeding.

Reviewer #2:

Remarks to the Author:

I would like to thank the authors for their detailed responses and explanations to my earlier comments. Their efforts in adding new sections of text, data, and assembly statistics are truly appreciated. However, I feel that the issue in my comment on Fig. 2f and Fig. 2g has not been fully addressed.

The synteny analysis in Fig. 2f shows major structural rearrangements across chromosome 6, which are most obvious between ~5 to ~45 Mbp. The authors use this figure as evidence that "confirmed the introgression of *Meloidogyne incognita* (Mi) resistance in Funtelle haplotype-1". To use this panel for their statement, the authors should clearly indicate the position of the Mi-1 locus in the synteny plots. Without the knowledge about the coordinates of Mi-1, the Fig. 2f just shows SVs between 4 haplotypes (and it remains unclear how the SVs relate to Mi-1). The same issue persists with the gene count analyses in Fig. 2g, which show that specific regions on chromosome 6 and 9 contain a high number of introgressed genes. To relate the regions on chr.6 with Mi-1 (and on chr.9 with Tm-2), the authors should clearly indicate the position of the respective genes in their plots.

Reviewer #3:

Remarks to the Author:

I have no further comments and I am satisfied with the revisions.

Reviewer #4:

Remarks to the Author:

In the revised manuscripts (NG-BC62972R1), the authors correctly and precisely responded to my concerns to the previous version of the manuscript, and the present manuscript will be suitable for publication.

Author Rebuttal, first revision:

Reviewers' Comments:

Reviewer #1:

Remarks to the Author:

The authors have addressed all my comments satisfactorily. There is just one point which needs elaboration in the manuscript: The estimated frequencies of unreduced female gametes in *TAM* mutants have been provided in the Rebuttal, but I could not find numbers for these estimates in the manuscript. They should be explicitly stated in the Supplementary Note 1, in the paragraph following the sentence "We also measured unreduced female gamete frequency by inference by performing crossing experiments using the *Sltam-3* and *Sltam-4* mutants." Other than this, the manuscript can be accepted without further revisions. It represents a significant contribution of high interest to plant genetics and breeding.

Response: Thank you very much for your comments! As suggested we have updated the Supplementary Note 1 by including the text we had previously provided in the point-by-point response. This is the text that has been added to the Supplementary Note 1:

"In summary, if we attribute all embryo outcomes that are not viable & diploid to the presence of an unreduced female gamete then we could estimate that in *Sltam-3* (70/153) 46% and in *Sltam-4* (111/201) 55% of female gametes are unreduced. More cautiously, some of the failed embryo development could be due to reasons other than the triploid block so a conservative estimate may suggest a female unreduced gamete frequency of between 30-50%. This number would be consistent with estimates for unreduced male gametes where all five *tam* lines produced pollen where 29-48% of pollen was unreduced (Fig.1c). Further corroboration of these numbers is found from the ploidy of *Sltam* mutant selfing offspring. We found that from *Sltam-3* selfing seed we retrieved 5 tetraploid plants out of 73 offspring, and for *Sltam-4* we retrieved 7 tetraploid plants out of 66 offspring (Extended Data Fig.1f). This is a cumulative tetraploidy rate of 9.3% which could be explained by 30% male unreduced gametes and 30% female unreduced gametes ($0.3 \times 0.3 \times 100 = 9\%$)."

Reviewer #2:

Remarks to the Author:

I would like to thank the authors for their detailed responses and explanations to my earlier comments. Their efforts in adding new sections of text, data, and assembly statistics are truly appreciated. However, I feel that the issue in my comment on Fig. 2f and Fig. 2g has not been fully addressed.

The synteny analysis in Fig. 2f shows major structural rearrangements across chromosome 6, which are most obvious between ~5 to ~45 Mbp. The authors use this figure as evidence that "confirmed the introgression of *Meloidogyne incognita* (*Mi*) resistance in Funtelle haplotype-1". To use this panel for their statement, the authors should clearly indicate the position of the *Mi-1* locus in the synteny plots. Without the knowledge about the coordinates of *Mi-1*, the Fig. 2f just shows SVs between 4 haplotypes (and it remains unclear how the SVs relate to *Mi-1*). The same issue persists with the gene count analyses in Fig. 2g, which show that specific regions on chromosome 6 and 9 contain a high number of introgressed genes. To relate the regions on chr.6 with *Mi-1* (and on chr.9 with *Tm-2*), the authors should clearly indicate the position of the respective genes in their plots.

Response: Thank you very much for your comments! We have improved the relevant figures (Fig. 2f and Fig. 2g) by clearly indicating the position of the *Mi-1* locus in the synteny plots (Fig. 2f) and *Mi-1* locus/*Tm-2* locus in Fig. 2g (see below). We have also clarified the main text as follows:

"Synteny analysis of the Funtelle haplotypes and annotation with a *Meloidogyne incognita* (*Mi-1*) resistance gene marker confirmed introgression in the structurally divergent Funtelle haplotype-1 (Fig.2f and Supplementary Fig.32), which was further elaborated by elevated numbers of *S. peruvianum* (*Mi-1* donor) genes along that haplotype (Fig.2g). The genotyping analysis showed that MbTMV-MT-Maxeza^{MiMe} plants have more copies of the *Tm-2*"

haplotype than MbTMV-MT-Funtelle^{MiMe} (Fig.2e and Supplementary Fig.32), as predicted from the parental genome sequences and distribution of *S. peruvianum* (*Tm-2²* donor) genes (Fig.2g and Supplementary Fig.33)."

Fig.2f, Genomic rearrangements on Chromosome 6 at the *Meloidogyne incognita* (*Mi-1*) resistance locus. The haplotypes are depicted as horizontal lines and are (from top to bottom) MbTMV, Micro-Tom, Funtelle-2 haplotype and Funtelle-1 haplotype.

Fig.2g, Frequency of genes derived from *S. peruvianum* (donor of *Tm-2²* and *Mi-1*) per genome haplotype

Reviewer #3:
Remarks to the Author:

I have no further comments and I am satisfied with the revisions.

Response: Thank you very much for your comments!

Reviewer #4:

Remarks to the Author:

In the revised manuscripts (NG-BC62972R1), the authors correctly and precisely responded to my concerns to the previous version of the manuscript, and the present manuscript will be suitable for publication.

Response: Thank you very much for your comments!

Decision Letter, second revision:

29th Feb 2024

Dear Dr. Underwood,

Thank you for submitting your revised manuscript "Harnessing clonal gametes in hybrid crops to engineer polyploid genomes" (NG-BC62972R2). The reviewers find that the paper has improved in revision, and therefore we'll be happy in principle to publish it in Nature Genetics, pending minor revisions to comply with our editorial and formatting guidelines.

Sincerely,
Wei

Wei Li, PhD
Senior Editor
Nature Genetics
New York, NY 10004, USA
www.nature.com/ng

Final Decision Letter:

9th Apr 2024

Dear Dr. Underwood,

I am delighted to say that your manuscript "Harnessing clonal gametes in hybrid crops to engineer polyploid genomes" has been accepted for publication in an upcoming issue of Nature Genetics.

Your paper will be published online after we receive your corrections and will appear in print in the next available issue. You can find out your date of online publication by contacting the Nature Press Office (press@nature.com) after sending your e-proof corrections.

Please note that *Nature Genetics* is a Transformative Journal (TJ). Authors may publish their research with us through the traditional subscription access route or make their paper immediately open access through payment of an article-processing charge (APC). Authors will not be required to make a final decision about access to their article until it has been accepted. Find out more about Transformative Journals

Authors may need to take specific actions to achieve compliance with funder and institutional open access mandates. If your research is supported by a funder that requires

immediate open access (e.g. according to Plan S principles) then you should select the gold OA route, and we will direct you to the compliant route where possible. For authors selecting the subscription publication route, the journal's standard licensing terms will need to be accepted, including <https://www.nature.com/nature-portfolio/editorial-policies/self-archiving-and-license-to-publish>. Those licensing terms will supersede any other terms that the author or any third party may assert apply to any version of the manuscript.

If you have not already done so, we invite you to upload the step-by-step protocols used in this manuscript to the Protocols Exchange, part of our on-line web resource, natureprotocols.com. If you complete the upload by the time you receive your manuscript proofs, we can insert links in your article that lead directly to the protocol details. Your protocol will be made freely available upon publication of your paper. By participating in natureprotocols.com, you are enabling researchers to more readily reproduce or adapt the methodology you use. [Natureprotocols.com](http://natureprotocols.com) is fully searchable, providing your protocols and paper with increased utility and visibility. Please submit your protocol to <https://protocolexchange.researchsquare.com/>. After entering your nature.com username and password you will need to enter your manuscript number (NG-BC62972R3). Further information can be found at <https://www.nature.com/nature-portfolio/editorial-policies/reporting-standards#protocols>

Sincerely,
Wei

Wei Li, PhD
Senior Editor
Nature Genetics

New York, NY 10004, USA
www.nature.com/ng